# NeuroRenderedFake: A Challenging Benchmark to Detect Fake Images Generated by Advanced Neural Rendering Methods

**Chengdong Dong**[1,2], **Vijayakumar Bhagavatula**[1], **Zhenyu Zhou**[2], **Ajay Kumar**[2]*
[1]Department of Electrical and Computer Engineering, Carnegie Mellon University
[2]Department of Data Science and Artificial Intelligence, The Hong Kong Polytechnic University
`chengdong.dong@connect.polyu.hk,kumar@ece.cmu.edu`
`zhenyucs.zhou@connect.polyu.hk,ajay.kumar@polyu.edu.hk`

## Abstract

The remarkable progress in neural-network-driven visual data generation, especially with neural rendering techniques like Neural Radiance Fields and 3D Gaussian splatting, offers a powerful alternative to GANs and diffusion models. These methods can generate high-fidelity images and lifelike avatars, highlighting the need for robust detection methods. However, the lack of *any* large dataset containing images from neural rendering methods becomes a bottleneck for the detection of such sophisticated fake images. To address this limitation, we introduce NeuroRendered-Fake, a comprehensive benchmark for evaluating emerging fake image detection methods. Our key contributions are threefold: (1) A large-scale dataset of fake images synthesized using state-of-the-art neural rendering techniques, significantly expanding the scope of fake image detection beyond generative models; (2) A cross-domain evaluation protocol designed to assess the domain gap and common artifacts between generative and neural rendering-based fake images; and (3) An in-depth spectral energy analysis that reveals how frequency domain characteristics influence the performance of fake image detectors. We train representative detectors, based on spatial, spectral, and multimodal architectures, on fake images generated by both generative and neural rendering models. We evaluate these detectors on 15 groups of fake images synthesized by cutting-edge neural rendering models, generative models, and *combined* methods that can exhibit artifacts from both domains. Additionally, we provide insightful findings through detailed experiments on degraded fake image detection and the impact of spectral features, aiming to advance research in this critical area.

## 1   Introduction

Images synthesized by generative models, such as Generative Adversarial Networks (GANs) [56, 57, 58] and Diffusion Models (DM) [59, 60, 61, 62], have raised significant ethical, privacy and security related concerns in our society. As noted in [52], the recursive generation of data can lead to model collapse during training, primarily due to contamination from the widespread availability of fake images. To address such concerns emerging from the generative models, numerous neural synthetic image detection models [1, 26, 31, 41, 25, 42, 86] have been developed.

However, the advancement of neural rendering technologies such as Neural Radiance Fields (NeRF) [9, 10, 2, 8, 11, 14, 17] and 3D Gaussian Splatting (3DGS) [7, 12, 20] offers a novel approach to generating highly realistic imagery such as scenes and digital humans/avatars, by the acquisition

---

*Corresponding author.

39th Conference on Neural Information Processing Systems (NeurIPS 2025) Track on Datasets and Benchmarks.

of two-dimensional projections from lifelike three-dimensional spatial representations. There even exist methodologies [6, 23] capable of directly editing the content within 3D representations. Unlike generative models, neural rendering technologies produce more realistic synthetic images by reconstructing scenes from actual images, thereby avoiding logical and semantic inconsistencies. This process allows for subtle 3D modifications that, when projected to 2D, are nearly imperceptible. Notably, current fake image detection systems have not addressed whether neural-rendered images can also be identified as *non-real*. This limitation prompts the question of whether current fake detectors possess sufficient generalization capabilities to detect neural-rendered images as fake.

Table 1: Comparative summary of related fake images dataset.

| Benchmark | Task Type | Image Synthesis Method | | | | combined | *real:fake* | releasing year |
|---|---|---|---|---|---|---|---|---|
| | | generative model | | neural rendering | | | | |
| | | GAN | DM | NeRF | 3DGS | | | |
| DFFD [64] | deepfake face | ✓ | ✗ | ✗ | ✗ | ✗ | 58,703: 240,336 | 2020 |
| ForgeryNet [65] | deepfake face | ✓ | ✗ | ✗ | ✗ | ✗ | 1,438,201: 1,457,861 | 2021 |
| DeepArt [66] | deepfake art | ✗ | ✓ | ✗ | ✗ | ✗ | 64,479: 73,411 | 2023 |
| CNNSpot [1] | general | ✓ | ✗ | ✗ | ✗ | ✗ | 362,000: 362,000 | 2020 |
| CIFAKE [67] | general | ✗ | ✓ | ✗ | ✗ | ✗ | 60,000: 60,000 | 2023 |
| UniFD [26] | general | ✗ | ✓ | ✗ | ✗ | ✗ | 1000: 8000 | 2023 |
| GenImage [68] | general | ✓ | ✓ | ✗ | ✗ | ✗ | 1,331,167: 1,350,000 | 2023 |
| Semi-Truth [84] | partial fake | ✗ | ✓ | ✗ | ✗ | ✗ | 27,600: 1,472,700 | 2024 |
| WildFake [78] | general | ✓ | ✓ | ✗ | ✗ | ✗ | 1,013,446: 2,557,278 | 2024 |
| *ours* | general+digital avatar | ✗ | ✓ | ✓ | ✓ | ✓ [1] | 512,972: 1,653,881 | 2025 |

[1] including GAN+NeRF, DM+NeRF, DM+3DGS for both generation and editing tasks

A major reason for this lack of evaluation is that the currently available benchmark datasets [64, 65, 66, 1, 67, 26, 68, 78] focus only on fake images synthesized by traditional generative models. To advance research on the detection of neural-rendered images, we have collected a large-scale database, which includes 296,504 images exclusively generated using a series of NeRF [9, 10, 2, 8, 11] and 3DGS [7, 2, 12] methods, 897,769 fake images by NeRF-based or 3DGS-based digital avatar synthesis methods [19, 20, 79], along with 287,258 images produced by approaches [13, 14, 16, 17, 6, 23] that *combine* generative models and neural rendering techniques, therefore introducing combined artifacts from both types of methods. Furthermore, to evaluate the cross-domain performance of fake image detectors, we also include in our database frames generated by state-of-the-art (SOTA) video generation models such as Sora [22], LivePortrait [77], and Runway [21], which demonstrate exceptional capability in generating consistent 3D scenes. Our database is made publicly available to encourage further research in this field. A comparative summary of our database and other related or widely-used fake image detection datasets is presented in Table 1. The previously popular databases listed in Table 1 do not include neural-rendered fake images. In contrast, our dataset offers a comprehensive collection of fake images generated by various types of neural rendering methods, including those combined techniques that integrate artifacts from both neural rendering and generative models. Notably, generating 3D scenes and projecting them onto a 2D view plane is computationally expensive and must be built from scratch. In this regard, the generation cost of our database, measured in GPU hours, is approximately three times higher than that of WildFake [78].

Additionally, we are interested in investigating whether fake images generated by neural rendering and generative models share common visual artifacts or exhibit domain-specific characteristics. Specifically, it remains unclear how well detectors trained exclusively on images generated by traditional generative models perform when tested on neural-rendered images—and vice versa. To address this issue, we provide a comprehensive evaluation protocol to assess the existence and extent of such domain gaps.

Furthermore, we introduce a multimodal architecture that leverages information from both the spatial and spectral domains, achieving superior performance compared to the current SOTA multimodal detector [42] on the test set of neural-rendered fake images. Our design involves pre-training the spatial and spectral branches independently, followed by a dynamic fusion mechanism for feature consolidation. The separate design of the spatial and spectral branches facilitates a clear assessment of the relative importance of features extracted from each domain.

In summary, the main contributions are: **1) A Large Dataset of Neural-rendered Fake Images:** Using NeRF- and 3DGS-based neural rendering techniques, we generate a variety of realistic 3D scene representations and render 2D fake images from various angles of view. Additionally, we synthesize fake images containing combined artifacts by integrating generative models with neural rendering technologies. In contrast to existing databases that focus exclusively on fake images from generative

models like GAN and DM, our diverse collection expands the scope of fake image detection, being the *first* in its inclusion of images derived from neural rendering-based synthesized/edited 3D scenes. **2) Comprehensive Cross-Domain Evaluations:** We conduct evaluations using representative detectors across multiple domains. Each detector is trained separately on fake images synthesized by generative models and those produced by neural rendering methods, and then tested on a wide range of fake image types. We establish a cross-domain evaluation protocol to identify both the common artifacts shared between generative and neural rendering models, as well as their domain-specific differences. **3) Analysis of Spectral Energy Distribution:** We analyze the contributions of spatial and spectral features to the final predictions in multimodal fake image detectors. Through an in-depth analysis of the spectral energy distributions of real and fake images, we uncover the relationship between frequency domain discrepancies and the inherent bias exhibited by multimodal detectors for the fake image detection.

## 2   Related Works

**Realistic 3D Scene Generation**: NeRF methods are developed to implicitly learn the 3D representation of specific scenes, which can be reconstructed from a series of input images [9, 10, 8, 11] or from a prompt such as a textual description [16] or a single-view image [69]. A potential malicious application involves integrating NeRF with editing techniques to alter the representation of 3D scenes using textual instructions [6, 36], images [13, 14], or both [35]. Additionally, NeRF-based methods are used for generating realistic digital humans (avatars), such as speech-to-video talking heads [27, 28, 18, 19, 79] and body synthesis [37, 69].

The 3DGS methods [7, 12] enable the learning of explicit 3D representations of scenes, which can be seamlessly integrated into existing rendering pipelines. Conditional editing techniques, such as Instruct-GS2GS [23], have been developed to modify 3DGS scenes. Avatar synthesis methods [29, 20, 79] based on 3DGS have also been proposed. Moreover, 3DGS can be combined with diffusion models to generate 3D scenes from scratch, as exemplified by GSGEN [17] and DreamGaussian [30]. Specifically, when neural rendering methods are integrated with generative models, such as approaches [13, 16, 14, 17, 6, 23], artifacts from both components can appear in the synthesized images.

**Fake Image Detection**: To detect images generated by generative models, traditional detectors based on the spatial domain [1, 33, 34, 71, 72, 70, 82, 80, 83] and those based on the spectral domain [25, 40, 41, 75, 74] are trained on real and fake images to identify the latent fingerprints of GANs and Deepfakes. Methods that require learning [32, 49, 50], and a learning-free method [47], exploit inherent properties of DM architecture for detecting DM-generated fakes. Several approaches [45, 48, 43, 26, 31, 42, 44] are effective in identifying both GAN- and DM-generated images. Methods [45, 48] enhance detection through an attention mechanism in the spectral domain. [44] utilizes captions from real images to generate fakes and then trains an SVM using deep features from both real and generated fakes. NPR [43] develops an operator to reveal neighboring pixel relationships within the spatial domain for improved detection. Ojha *et al.* [26] introduce a large vision model with a fixed backbone to improve generalization, while [42, 31] fine-tune the backbone of large vision models like LoRA [63] to enhance representation while maintaining generalization. FatFormer [42] further integrates frequency and language cues to improve cross-domain generalization, while [85] introduces a semantic-guided module to detect diffusion-generated hyper-realistic videos.

## 3   Introduction to New Dataset and Organization

Based on the aforementioned discussion, we primarily classify neural-rendered fake images into three categories: (1) fake images generated exclusively by neural rendering methods, (2) fake images containing combined artifacts from both neural rendering techniques and generative models, and (3) a popular research topic—digital avatars generated using neural rendering methods. Additionally, to evaluate detectors trained on fake images with neural-rendered artifacts but tested on currently popular generative models, we also collect a large number of fake images produced by video synthesis methods. The development of this new database (Table 1) is detailed in the following section and is made publicly accessible via [87] to advance much-needed research in this area.

**Exclusive Neural Rendered Images:** We have acquired 139 groups of consecutive 2D images, each accompanied by the corresponding camera poses details. The information of those images are

Table 2: The details of generated scenes and the corresponding 2D images of exclusive neural rendered images. Numbers before and after / denote the number of generated 3D scenes and rendered 2D images, respectively. Only 3D scenes that are successfully reconstructed are used.

| Sources of the Scenes | real | I [9] i-ngp | II [10] tensorf | III [2] nerfacto | IV [8] seaThru | V [11] pynerf | VI [7] 3dgs | VII [2] splatfacto | VIII [12] C3dgs |
|---|---|---|---|---|---|---|---|---|---|
| **A** blender [2] | 8/800 | 8/800 | 8/800 | 6/600 | 8/800 | 8/800 | ✗ | 2/200 | ✗ |
| **B** D-NeRF [2] | 8/1,100 | 7/1,000 | 8/1,100 | 5/550 | 8/1,100 | 8/1,100 | ✗ | ✗ | ✗ |
| **C** eyeful-T [2] | 11/28,572 | 11/28,572 | 8/25,107 | 11/28,572 | 8/15,465 | 5/10,668 | ✗ | 7/9,654 | ✗ |
| **D** mill19 [2] | 2/3,618 | 2/3,618 | 2/3,618 | 2/3,618 | 2/3,618 | 2/3,618 | ✗ | 2/3,618 | ✗ |
| **E** nerfosr [2] | 9/4,703 | 9/4,426 | 7/3,634 | 9/4,426 | ✗ | ✗ | ✗ | 7/3,634 | ✗ |
| **F** nerfstudio [2] | 17/6,321 | 17/6,321 | 17/6,321 | 17/6,321 | 17/6,321 | 11/4,605 | ✗ | 17/6,321 | ✗ |
| **G** phototour [2] | 10/17,741 | ✗ | ✗ | 10/17,741 | ✗ | ✗ | ✗ | 8/10,785 | 8/10,785 |
| **H** record3d [2] | 1/300 | 1/300 | 1/300 | 1/300 | 1/300 | 1/300 | ✗ | 1/300 | ✗ |
| **I** sdfStudio [2] | 34/4,771 | 34/4,771 | 34/4,771 | 34/4,771 | 32/4,172 | 31/3,871 | ✗ | 34/4,771 | ✗ |
| **J** sitcoms3d [2] | 10/1,451 | ✗ | 10/1,451 | ✗ | ✗ | ✗ | ✗ | 10/1,451 | ✗ |
| **K** mip360 [3] | 9/1,940 | 9/1,940 | 8/1,755 | 9/1,940 | 9/1,940 | 8/1,755 | 9/1,940 | 9/1,940 | 9/1,940 |
| **L** llff [4] | 8/305 | 7/250 | 2/45 | 8/305 | 4/105 | 5/146 | 8/305 | 8/305 | 8/305 |
| **M** head [6, 5] | 4/353 | 4/353 | 1/65 | 4/353 | 2/160 | 1/65 | 3/288 | 4/353 | 3/288 |
| **N** inria [7] | 4/1,040 | 2/488 | 2/488 | 2/488 | 2/488 | 2/488 | 4/1,040 | 2/488 | 4/1,040 |
| **O** underwater [8] | 4/88 | 4/88 | 1/20 | 4/88 | 4/88 | 2/41 | 4/88 | 4/88 | 4/88 |
| sum scenes/images | 139/73,103 | 115/52,927 | 109/49,475 | 122/70,073 | 97/34,557 | 84/27,457 | 28/3,661 | 115/43,908 | 36/14,446 |

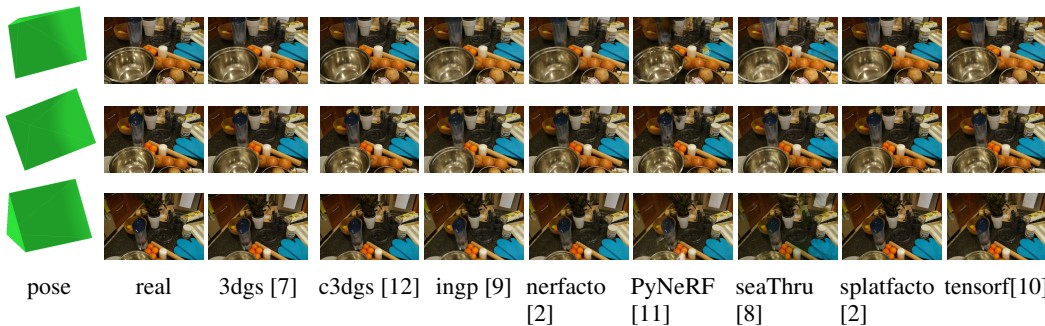

| pose | real | 3dgs [7] | c3dgs [12] | ingp [9] | nerfacto [2] | PyNeRF [11] | seaThru [8] | splatfacto [2] | tensorf[10] |

Figure 1: Samples of fake images in our dataset generated by exclusive neural rendering methods.

organized in Table 2. These camera poses were either recorded during the photo capture process or calibrated using structure-from-motion techniques. To generate *fake* images, we employed a variety of methods capable of producing realistic 3D representations from the aforementioned data. Specifically, we utilized different NeRF-based and 3DGS-based methods, labeled from **I** to **VIII**, to generate 3D scene representations. Once the 3D scenes are reconstructed, they are projected back to 2D images using the same projection parameters as those estimated from the original 2D photos. These rendered 2D images are also considered *fake* in this paper, as they inherently contain renderer-specific imprints that distinguish them from real images. In experiments, the original 2D photos used for reconstruction are regarded as *real* images. **Combined Artifacts:** We collect six widely used methods that generate combined fake artifacts through neural rendering and generative models. Pix2NeRF [13] and SketchFaceNeRF [14] are image-to-image generation methods that combine GANs with NeRF. DreamFusion [16], a text-to-image method, integrates DM with NeRF, while GSGEN [17] applies DM+3DGS. We also include i-N2N [6] and i-GS2GS [23], which are image-to-image editing methods based on DM+NeRF and DM+3DGS, respectively. **Neural Rendered Digital Avatar:** We

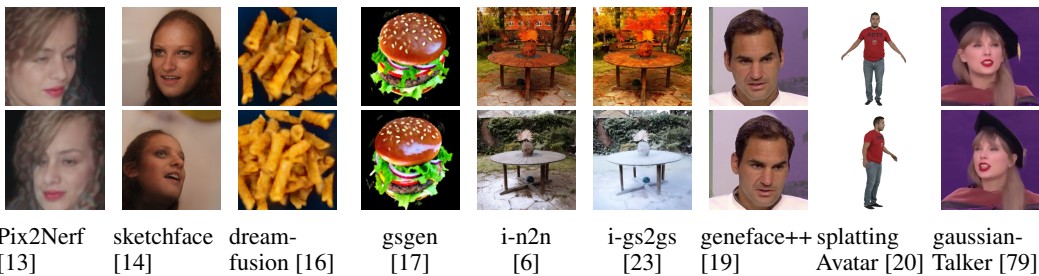

| Pix2Nerf [13] | sketchface [14] | dream-fusion [16] | gsgen [17] | i-n2n [6] | i-gs2gs [23] | geneface++ [19] | splatting Avatar [20] | gaussian-Talker [79] |

Figure 2: Samples in our dataset of fake images beyond exclusive neural rendering methods.

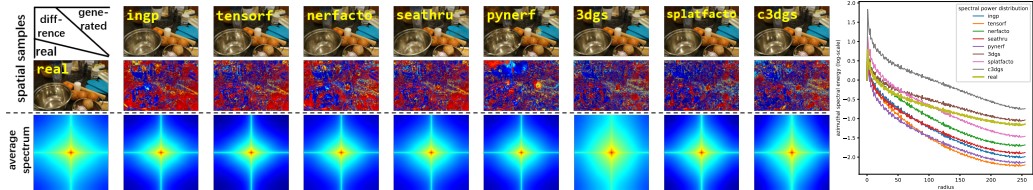

Figure 3: The averaged 2D magnitude of the spectrum and 1D spectral energy distribution of the real and neural rendered fake images are provided.

generate speech-driven digital avatar of Geneface++ [18] and GaussianTalker [79] based on NeRF and 3DGS, respectively, And we use splattingAvatar [20] for head/body synthesis based on 3DGS. **Realistic Video Generation:** We collect the video frames from the currently popular realistic video generation methods mainly based on diffusion models, such as Sora [22], Liveportrait [77] and Runway [21].

In Fig. 1, we visualize fake image samples of the exclusive neural rendered images, and their corresponding camera poses, all belonging to the same 3D scene. Similarly, images samples in Fig. 2 present fake image samples from the database for the performance evaluation, which are generated by a variety of synthesis methods, including editable NeRF, editable 3DGS, and combined methods of NeRF/3DGS + GAN/DM, extending beyond exclusive neural rendering approaches. In Fig. 3, we visualize samples of images belonging to the same scene, including both real images and neural-rendered fakes. We also illustrate the differences between real and fake images in the spatial domain, along with the averaged 2D magnitude of the spectrum and the 1D spectral energy distribution for both real and neural-rendered fake images. The observed discrepancies in both the spatial and spectral domains suggest the feasibility of detecting neural-rendered fake images. More details of dataset construction are presented in the Sec. C of the Appendix.

## 4 Experimental Results and Analysis

### 4.1 Design of Multimodal Architecture for Detecting NeuroRenderedFake

In some of the SOTA methods, e.g. [42], which are based on a multimodal architecture, the information extracted from the spatial and spectral domains is highly correlated, making it difficult to analyze the individual contribution of each domain to the final prediction. To investigate the respective contributions of the spatial and spectral domains, we propose a multimodal architecture that includes separate spatial and spectral branches for extracting information from each domain. These two branches remain independent until the final consolidation layer, and the contributions of the spatial and spectral features to the final output are explicitly modeled through learnable weights. We adopt a Transformer-based backbone comprising ViT-L14 blocks for the spatial branch and ViT-B16 blocks for the frequency branch, which we refer to as **F**ourier-**F**requency-**i**nformed **T**ransformer (FFiT). AdaLoRA [55] blocks are incorporated into both the spatial and spectral branches to enable dynamic fine-tuning of the pre-trained parameters. We first pre-train the two branches independently on the training dataset. Subsequently, we fix the pre-trained weights and employ a Gated Multimodal Unit (GMU) [51], as defined in Eq. (1), to perform information fusion.

$$w_a = l(f_a \odot f_b), \ w_b = 1 - w_a, \ f_{mm} = w_a f_a + w_b f_b \tag{1}$$

, where $l(\cdot)$ denotes the learnable hidden layers followed by a fully connected layer in the GMU module, with its output being a scalar. $f_a$ and $f_b$ represent the normalized features from the spatial and spectral branches, respectively, while $f_{mm}$ denotes the multimodal feature. $w_a$ and $w_b$ indicate the weights assigned to the contributions from each branch.

During this fusion stage, only the parameters within the GMU block are updated. Both individual branches and the overall multimodal architecture are evaluated in our experiments. Further details regarding the architectural design and training procedures are provided in Sec. A of the Appendix.

### 4.2 Task 1: Cross-Domain Evaluation

During the training and evaluation of the detectors, we resize the images to $256 \times 256$ and subsequently crop them to $224 \times 224$ pixels. For training the detectors, we utilize four distinct datasets $(\mathcal{A}, \mathcal{B}, \mathcal{C}, \mathcal{D})$,

each containing real images as well as fake images generated by GAN, DM, NeRF, and 3DGS, respectively. These datasets are used to train different fake detectors in our experiments. The detailed construction process of the datasets $\mathcal{A}, \mathcal{B}, \mathcal{C}, \mathcal{D}$ is provided in Sec. C of the Appendix.

Table 3: Evaluation dataset with number of real/fake

| ours' collection | | | other source |
|---|---|---|---|
| neural rendering | | generative model | generative model |
| 1.**I** $\sim$ **V** (unseen NeRF) 3,726/13,942 | 2.**VI** $\sim$ **VIII** (unseen 3DGS) 3,726/10,496 | 12.Sora [22] | 15.MM-Det [81] |
| 3.Pix2NeRF [13] (GAN+NeRF) 70,000/96,000 | 4.SketchFaceNeRF [14] (GAN+NeRF) 70,000/90,000 | (DM video) 60,000/60,531 | (3 methods) |
| 5.DreamFusion [16] (DM+NeRF) 10,000/10,600 | 6.GSGEN [17] (DM+3DGS) 10,000/9,540 | 13.Liveportrait [77] | 28,770/28,000 |
| 7.instruct-N2N [6] (DM+NeRF) 3,174/40,559 | 8.instruct-GS2GS [23] (DM+3DGS) 3,174/40,559 | (DM avatar)59,260/88,466 | 16.GenImage [68] |
| 9.GeneFace++ [19] (Avatar NeRF) 88,737/452,653 | 10.SplattingAvatar [20] (Avatar 3DGS) 33,728/33,715 | 14. Runway [21] | (8 methods) |
| 11.GaussianTalker [79] (Avatar 3DGS) 94,810/411,401 | | (DM video)23,334/23,353 | 50,000/50,000 |

Table 4: Comparative performances on detecting neural rendered fake images. (in %)

| | | | Average Precision | | | | | | | | | | | | AUROC | | | | | | | | | | | |
|---|---|---|---|---|---|---|---|---|---|---|---|---|---|---|---|---|---|---|---|---|---|---|---|---|---|---|
| | | group | **1** | **2** | **3** | **4** | **5** | **6** | **7** | **8** | **9** | **10** | **11** | ave | **1** | **2** | **3** | **4** | **5** | **6** | **7** | **8** | **9** | **10** | **11** | ave |
| spatial- based | CNN-Spot [1] | $\mathcal{A}$ | 83.20 | 76.19 | 89.75 | 95.66 | 67.04 | 56.14 | 98.11 | 98.73 | 90.06 | 68.33 | 90.67 | 83.08 | 56.02 | 52.16 | 91.47 | 95.04 | 69.66 | 64.34 | 78.60 | 84.76 | 65.18 | 74.25 | 71.17 | 72.97 |
| | | $\mathcal{B}$ | 88.03 | 75.21 | 88.49 | 97.29 | 84.20 | 93.21 | 98.05 | 98.49 | 85.12 | 49.08 | 78.00 | 85.02 | 66.04 | 52.75 | 88.39 | 96.94 | 86.78 | 94.68 | 81.07 | 85.27 | 55.53 | 43.59 | 46.49 | 72.50 |
| | | $\mathcal{C}$ | / | 87.06 | 72.89 | 80.47 | 70.36 | 82.55 | 99.97 | 99.79 | 85.84 | 84.62 | 84.84 | 84.83 | / | 66.67 | 73.15 | 79.92 | 64.83 | 79.33 | 99.65 | 97.61 | 53.83 | 92.05 | 55.40 | 76.24 |
| | | $\mathcal{D}$ | 94.31 | / | 92.80 | 65.68 | 61.84 | 96.44 | 99.36 | 99.95 | 77.56 | 73.96 | 75.09 | 83.70 | 80.77 | / | 92.53 | 64.92 | 59.24 | 94.46 | 92.74 | 99.46 | 41.99 | 84.99 | 41.52 | 75.26 |
| | | ave | 88.51 | 79.49 | 85.98 | 84.77 | 70.86 | 82.08 | 98.87 | 99.24 | 84.64 | 68.99 | 82.15 | | 67.61 | 57.19 | 86.38 | 84.20 | 70.12 | 83.20 | 88.01 | 91.77 | 54.13 | 73.72 | 53.65 | |
| | UniFD [26] | $\mathcal{A}$ | 93.82 | 77.02 | 92.78 | 92.46 | 61.63 | 73.15 | 96.09 | 97.72 | 86.65 | 61.19 | 87.25 | 83.61 | 80.56 | 55.90 | 91.39 | 91.44 | 61.55 | 77.40 | 72.49 | 84.41 | 58.79 | 57.79 | 62.36 | 72.19 |
| | | $\mathcal{B}$ | 94.19 | 88.26 | 76.57 | 99.31 | 93.66 | 93.07 | 99.39 | 99.79 | 89.60 | 71.77 | 89.57 | 90.47 | 82.21 | 73.77 | 67.68 | 99.03 | 92.94 | 93.55 | 94.00 | 97.97 | 63.47 | 70.60 | 67.21 | 82.04 |
| | | $\mathcal{C}$ | / | 83.65 | 75.23 | 82.14 | 76.50 | 66.72 | 99.89 | 99.91 | 89.25 | 72.55 | 88.30 | 83.41 | / | 65.56 | 71.17 | 81.80 | 71.98 | 73.78 | 98.83 | 99.02 | 59.60 | 70.60 | 61.49 | 75.38 |
| | | $\mathcal{D}$ | 95.67 | / | 92.09 | 86.97 | 73.83 | 62.63 | 99.82 | 99.97 | 88.57 | 83.61 | 87.98 | 87.11 | 86.02 | / | 90.08 | 85.73 | 71.04 | 67.37 | 97.92 | 99.72 | 63.09 | 82.91 | 67.76 | 81.16 |
| | | ave | 94.56 | 82.97 | 84.16 | 90.22 | 76.40 | 73.89 | 98.79 | 99.34 | 88.51 | 72.28 | 88.28 | | 82.93 | 65.07 | 80.08 | 89.50 | 74.37 | 78.02 | 90.81 | 95.28 | 61.24 | 70.47 | 64.71 | |
| | MoeFD [31] | $\mathcal{A}$ | 86.08 | 82.50 | 91.43 | 98.72 | 73.71 | 85.68 | 96.32 | 99.78 | 87.13 | 60.42 | 89.84 | 86.51 | 60.41 | 63.85 | 90.86 | 98.42 | 74.50 | 86.10 | 64.49 | 97.36 | 66.37 | 60.16 | 68.29 | 75.53 |
| | | $\mathcal{B}$ | 87.93 | 85.03 | 88.65 | 99.85 | 98.94 | 94.94 | 98.22 | 99.26 | 90.56 | 65.32 | 85.62 | 90.39 | 65.73 | 69.70 | 88.80 | 99.80 | 98.82 | 95.05 | 81.87 | 91.62 | 64.18 | 63.93 | 65.74 | 80.48 |
| | | $\mathcal{C}$ | / | 87.32 | 91.20 | 88.96 | 82.23 | 78.09 | 96.77 | 99.82 | 90.98 | 67.69 | 89.66 | 87.27 | / | 74.94 | 90.66 | 89.34 | 82.87 | 78.76 | 68.28 | 97.86 | 67.41 | 65.77 | 69.48 | 78.54 |
| | | $\mathcal{D}$ | 89.68 | / | 90.13 | 90.46 | 78.20 | 88.69 | 98.13 | 99.49 | 89.75 | 71.25 | 90.23 | 88.60 | 70.94 | / | 89.86 | 90.63 | 78.94 | 89.01 | 80.97 | 94.11 | 65.09 | 68.62 | 71.51 | 79.97 |
| | | ave | 87.89 | 84.95 | 90.35 | 94.49 | 83.27 | 86.85 | 97.36 | 99.58 | 89.60 | 66.17 | 88.84 | | 65.69 | 69.49 | 90.04 | 94.54 | 83.78 | 87.23 | 73.90 | 95.23 | 65.76 | 64.62 | 68.76 | |
| | RINE [80] | $\mathcal{A}$ | 92.58 | 92.30 | 90.44 | 95.83 | 98.97 | 97.36 | 99.02 | 98.36 | 89.62 | 62.29 | 88.21 | 91.36 | 87.13 | 87.56 | 86.90 | 92.72 | 94.66 | 92.91 | 97.88 | 97.59 | 69.50 | 61.59 | 67.16 | 85.05 |
| | | $\mathcal{B}$ | 91.67 | 90.56 | 87.20 | 96.89 | 99.05 | 98.74 | 97.52 | 99.14 | 92.01 | 69.13 | 86.04 | 91.63 | 88.50 | 86.92 | 85.41 | 93.49 | 97.06 | 94.64 | 96.13 | 98.95 | 61.34 | 68.38 | 65.88 | 85.15 |
| | | $\mathcal{C}$ | / | 91.20 | 90.84 | 95.60 | 98.47 | 97.65 | 99.24 | 98.63 | 90.82 | 60.41 | 89.15 | 91.20 | / | 87.78 | 87.57 | 93.84 | 95.66 | 93.96 | 98.61 | 98.53 | 65.67 | 62.85 | 70.40 | 85.49 |
| | | $\mathcal{D}$ | 94.23 | / | 91.10 | 98.52 | 98.28 | 97.57 | 99.18 | 99.25 | 91.86 | 70.48 | 92.64 | 93.31 | 83.74 | / | 89.17 | 98.60 | 95.44 | 98.54 | 98.98 | 97.67 | 62.97 | 83.22 | 75.27 | 88.36 |
| | | ave | 92.83 | 91.35 | 89.90 | 96.71 | 98.69 | 97.83 | 98.74 | 98.85 | 91.08 | 65.58 | 89.01 | | 86.46 | 87.42 | 87.26 | 94.66 | 95.71 | 95.01 | 97.90 | 98.19 | 64.87 | 69.01 | 69.58 | |
| | RGB branch (ours) | $\mathcal{A}$ | 93.38 | 92.02 | 78.57 | 91.25 | 98.95 | 98.12 | 97.76 | 97.98 | 91.84 | 49.43 | 92.29 | 89.24 | 85.38 | 85.15 | 76.32 | 93.47 | 99.20 | 98.52 | 83.25 | 83.53 | 70.42 | 54.81 | 76.76 | 82.44 |
| | | $\mathcal{B}$ | 87.67 | 83.77 | 90.95 | 98.51 | 99.54 | 99.14 | 98.35 | 99.81 | 91.83 | 59.04 | 91.01 | 90.87 | 68.88 | 68.17 | 89.52 | 98.37 | 99.60 | 99.27 | 84.12 | 97.76 | 68.96 | 57.85 | 71.16 | 82.15 |
| | | $\mathcal{C}$ | / | 99.25 | 86.01 | 94.57 | 99.33 | 98.16 | 99.83 | 99.87 | 92.62 | 53.31 | 92.75 | 91.57 | / | 98.79 | 84.82 | 95.95 | 99.49 | 98.62 | 98.70 | 98.99 | 72.97 | 58.71 | 77.48 | 88.45 |
| | | $\mathcal{D}$ | 99.45 | / | 92.99 | 99.76 | 99.94 | 99.62 | 99.98 | 99.99 | 94.42 | 77.88 | 95.21 | 95.92 | 98.05 | / | 90.76 | 99.71 | 99.94 | 99.68 | 99.84 | 99.96 | 77.28 | 76.06 | 83.00 | 92.43 |
| | | ave | 93.50 | 91.68 | 87.13 | 96.02 | 99.44 | 98.76 | 98.98 | 99.41 | 92.67 | 59.92 | 92.82 | | 84.10 | 84.04 | 85.36 | 96.88 | 99.56 | 99.02 | 91.48 | 95.06 | 72.40 | 61.86 | 77.10 | |
| spectral- based | Freq-spec [25] | $\mathcal{A}$ | 81.83 | 74.59 | 94.94 | 79.52 | 79.19 | 77.15 | 96.64 | 97.69 | 86.20 | 67.29 | 86.00 | 83.73 | 53.15 | 49.77 | 96.87 | 81.84 | 82.70 | 84.86 | 70.39 | 78.00 | 61.87 | 68.45 | 66.02 | 72.17 |
| | | $\mathcal{B}$ | 83.15 | 76.44 | 84.47 | 79.26 | 80.74 | 76.85 | 96.58 | 96.88 | 87.10 | 70.78 | 82.05 | 83.12 | 58.51 | 54.48 | 87.94 | 80.60 | 81.49 | 85.83 | 71.68 | 73.74 | 59.67 | 74.74 | 55.72 | 71.30 |
| | | $\mathcal{C}$ | / | 87.48 | 97.07 | 90.64 | 59.34 | 92.39 | 99.06 | 99.05 | 87.20 | 64.83 | 85.59 | 86.29 | / | 70.42 | 96.18 | 84.73 | 46.72 | 94.21 | 93.10 | 93.33 | 59.44 | 76.46 | 58.07 | 77.28 |
| | | $\mathcal{D}$ | 96.60 | / | 89.56 | 81.52 | 54.01 | 90.47 | 99.71 | 99.68 | 88.12 | 86.79 | 91.37 | 87.79 | 89.70 | / | 91.27 | 68.74 | 65.54 | 93.68 | 97.17 | 96.86 | 61.53 | 86.37 | 72.92 | 82.42 |
| | | ave | 87.19 | 79.50 | 91.51 | 82.73 | 68.32 | 84.22 | 98.00 | 98.33 | 87.16 | 72.42 | 86.25 | | 67.12 | 58.22 | 93.06 | 78.98 | 69.11 | 89.64 | 83.08 | 85.48 | 60.63 | 76.51 | 63.18 | |
| | FFiT (ours) | $\mathcal{A}$ | 89.55 | 82.41 | 99.96 | 96.19 | 97.75 | 97.19 | 97.78 | 98.08 | 89.98 | 81.10 | 88.42 | 92.58 | 69.86 | 62.46 | 99.95 | 95.17 | 97.67 | 97.98 | 78.67 | 81.82 | 64.03 | 81.76 | 64.03 | 81.22 |
| | | $\mathcal{B}$ | 97.67 | 90.98 | 95.54 | 98.73 | 90.30 | 93.95 | 99.00 | 99.12 | 91.67 | 55.15 | 89.67 | 91.07 | 92.30 | 80.30 | 92.94 | 98.94 | 90.04 | 95.21 | 88.95 | 90.53 | 71.96 | 50.42 | 70.06 | 83.79 |
| | | $\mathcal{C}$ | / | 82.66 | 97.90 | 99.91 | 78.53 | 72.24 | 97.67 | 97.59 | 89.99 | 75.82 | 84.52 | 87.68 | / | 64.04 | 98.06 | 99.87 | 78.51 | 79.04 | 81.11 | 80.52 | 64.99 | 73.27 | 55.26 | 77.47 |
| | | $\mathcal{D}$ | 92.75 | / | 99.11 | 94.38 | 64.26 | 96.77 | 99.47 | 99.66 | 84.24 | 96.48 | 81.02 | 90.81 | 79.06 | / | 98.84 | 92.77 | 54.91 | 95.81 | 94.62 | 96.95 | 51.39 | 96.76 | 48.79 | 80.99 |
| | | ave | 93.32 | 85.35 | 98.13 | 97.30 | 82.71 | 90.04 | 98.48 | 98.61 | 88.97 | 77.14 | 85.91 | | 80.41 | 68.93 | 97.45 | 96.69 | 80.28 | 92.01 | 85.84 | 87.46 | 63.09 | 75.55 | 59.54 | |
| multimodal | Fat-Former [42] | $\mathcal{A}$ | 95.76 | 94.93 | 99.37 | 99.65 | 99.35 | 99.08 | 99.38 | 99.50 | 91.38 | 80.16 | 92.21 | 95.52 | 86.89 | 87.93 | 99.30 | 99.55 | 99.40 | 99.23 | 93.22 | 94.54 | 72.43 | 82.07 | 74.76 | 89.94 |
| | | $\mathcal{B}$ | 93.28 | 87.40 | 99.70 | 99.99 | 99.83 | 99.87 | 98.85 | 99.92 | 92.45 | 76.35 | 91.68 | 94.48 | 81.56 | 75.09 | 99.60 | 99.99 | 99.84 | 99.89 | 88.35 | 99.02 | 72.65 | 75.13 | 72.50 | 87.60 |
| | | $\mathcal{C}$ | / | 94.25 | 99.88 | 99.99 | 98.87 | 97.36 | 99.86 | 99.90 | 91.69 | 82.72 | 91.97 | 95.65 | / | 85.59 | 99.85 | 99.99 | 98.43 | 97.30 | 98.32 | 98.93 | 71.47 | 83.13 | 70.54 | 90.36 |
| | | $\mathcal{D}$ | 96.16 | / | 99.76 | 99.78 | 98.32 | 99.74 | 99.89 | 99.66 | 92.30 | 97.44 | 91.40 | 97.45 | 86.90 | / | 99.66 | 99.70 | 97.66 | 99.72 | 98.56 | 96.30 | 70.95 | 97.59 | 72.39 | 91.94 |
| | | ave | 95.07 | 92.47 | 99.68 | 99.85 | 99.09 | 99.01 | 99.50 | 99.75 | 91.96 | 84.17 | 91.82 | | 85.12 | 82.87 | 99.60 | 99.81 | 98.83 | 99.04 | 94.61 | 97.20 | 71.87 | 84.48 | 72.55 | |
| | Ours | $\mathcal{A}$ | 97.56 | 96.21 | 99.15 | 99.86 | 99.79 | 99.56 | 99.68 | 99.83 | 92.91 | 83.38 | 93.43 | 96.49 | 91.76 | 90.67 | 99.08 | 99.82 | 99.80 | 99.62 | 96.28 | 98.00 | 72.19 | 84.89 | 77.31 | 91.77 |
| | | $\mathcal{B}$ | 91.82 | 87.86 | 99.78 | 99.99 | 99.96 | 99.98 | 98.99 | 99.93 | 92.70 | 73.15 | 92.52 | 94.24 | 77.52 | 76.09 | 99.70 | 99.99 | 99.96 | 99.98 | 89.73 | 99.21 | 72.41 | 71.83 | 74.56 | 87.36 |
| | | $\mathcal{C}$ | / | 93.79 | 99.75 | 99.99 | 99.27 | 98.40 | 99.76 | 99.83 | 92.83 | 84.71 | 91.64 | 96.00 | / | 84.49 | 99.69 | 99.99 | 98.98 | 98.39 | 97.33 | 98.18 | 72.96 | 84.94 | 71.86 | 90.68 |
| | | $\mathcal{D}$ | 97.98 | / | 99.89 | 99.91 | 97.03 | 99.94 | 99.92 | 99.97 | 91.56 | 99.18 | 91.83 | 97.72 | 92.96 | / | 99.84 | 99.88 | 95.59 | 99.94 | 99.10 | 99.74 | 69.09 | 99.09 | 73.52 | 92.88 |
| | | ave | 95.79 | 92.62 | 99.64 | 99.94 | 99.01 | 99.47 | 99.59 | 99.89 | 92.50 | 85.11 | 92.36 | | 87.41 | 83.75 | 99.58 | 99.92 | 98.58 | 99.48 | 95.61 | 98.78 | 71.66 | 85.19 | 74.31 | |

**Cross-domain Evaluation on Neural Rendered Fake Images:** In Table 3, we list the groups of testing dataset for evaluating the detectors' performance, and the group numbers range from 1 to 11 are all fake images generated by neural rendering methods. In Table 4, the cross-domain testing performance for different detectors trained on groups $\mathcal{A}$, $\mathcal{B}$, $\mathcal{C}$, and $\mathcal{D}$ and tested on groups 1 to 11 is provided. During re-implentation of the baseline detectors, we follow the same parameter setting for the baseline detectors as mentioned in their papers. The performance evaluated on group $1 \sim 11$ from the proposed method is compared with re-implemented detectors [1, 26, 31, 80] for the spatial domain only, and with the re-implemented detector [25] for the spectral domain only. The multimodal backbone is compared with FatFormer [42] by re-implementing the spatial-spectral multimodal branch of [42] while removing the language-based branch for fair comparison. This adjustment not only streamlines the comparison but also mitigates discrepancies in the text-guided interaction [42] module during evaluation, given the partial and low-level scenes prevalent in the NeRF/3DGS-generated images of the testing dataset, unlike the uniform rich contexts of GAN/DM outputs used for testing in [42].

For a rigorous cross-domain evaluation, the performance metrics of detectors that are both trained and tested on images generated by the same method, such as those synthesized by NeRF and tested on NeRF (group $\mathcal{C} - \mathbf{1}$) or synthesized by 3DGS and tested on 3DGS (group $\mathcal{D} - \mathbf{2}$), are not included

in the Table 4. These cases are considered within-domain evaluations and do not align with the cross-domain assessment objectives. Instead, the findings from these within-domain tests are described in the cross-time evaluation in the Section 4.3, adhering to a separate evaluation protocol.

The table also includes the average AP and average AUROC for each of the four training groups on the 11 testing groups, as well as the average AP and average AUROC for each of the 11 testing groups on the four training groups. From Table 4, it can be concluded that the method performs well when our developed spatial and spectral branches operate independently. Additionally, the design of the multimodal backbone demonstrates superior performance in detecting fake images generated by different methods across testing groups 1 to 11, compared to other SOTA fake detectors in the spatial domain, spectral domain, and multimodal domain.

**Cross-domain Evaluation on Fake Image Dataset of Generative Models:** To comprehensively assess the discrepancy between artifacts left by generative models and neural rendering models, we evaluated the performance of various detectors trained on groups $\mathcal{A}$, $\mathcal{B}$, $\mathcal{C}$, and $\mathcal{D}$ against fake images produced by advanced generative models, as shown in Table 5 with the AUROC values provided. From Table 5, it is evident that detectors pretrained on neural rendered fake images (groups $\mathcal{C}$ and $\mathcal{D}$) are capable of detecting fake images generated by advanced generative models; however, their performance does not consistently surpass those pretrained on groups $\mathcal{A}$ and $\mathcal{B}$. Notably, our analysis reveals that the best detection performance, indicated by the highest AUROC value, is achieved by our multimodal detector trained on group $\mathcal{D}$. This suggests that the fake images generated by 3DGS have a small domain gap with the testing datasets of generative models listed in Table 5. From the results obtained using the spectral-based detector, it is observed that its performance is poor in detecting fake images generated by advanced generative models. This negatively impacts the judgment of the spatial domain when combined into a multimodal detector. Specifically, several groups of performance of our multimodal detector falls between that of the detector using our RGB branch alone and the FFiT branch alone. This indicates that the addition of the spectral-based detector does not consistently improve detection performance and can sometimes degrade overall results when used in conjunction with spatial domain detectors. Furthermore, the Runway dataset [21] poses a significant challenge for fake detection tasks, as none of the detectors achieved satisfactory results on this dataset.

Table 5: Generalization results on fake image dataset of traditional generative models evaluated by AUROC metric in %

| Method | video (ours) | | | 15. video of MM-Det [81] | | | 16. GenImage [68] fake dataset of generative models | | | | Stable Diffusion | | | | Total |
| | 12. sora | 13. live-portrait | 14. runway | open-sora | video-crafter | zero-scope | ADM | Big-GAN | Glide | Mid-Journey | v1.4 | v1.5 | VQDM | wukong | average |
| --- | --- | --- | --- | --- | --- | --- | --- | --- | --- | --- | --- | --- | --- | --- | --- |
| CNNSpot [1] $\mathcal{A}$ | 39.21 | 68.94 | 55.74 | 63.93 | 79.35 | 77.54 | 72.03 | 90.78 | 92.78 | 45.86 | 72.72 | 73.11 | 72.35 | 72.00 | 69.74 |
| CNNSpot [1] $\mathcal{B}$ | 60.62 | 58.38 | 48.06 | 54.89 | 64.00 | 67.71 | 71.60 | 99.00 | 78.43 | 36.64 | 59.87 | 59.65 | 93.84 | 55.32 | 64.86 |
| CNNSpot [1] $\mathcal{C}$ | 68.19 | 49.92 | 59.65 | 66.24 | 51.97 | 75.22 | 74.64 | 81.57 | 72.46 | 66.18 | 72.14 | 72.27 | 66.97 | 54.44 | 66.56 |
| CNNSpot [1] $\mathcal{D}$ | 55.47 | 49.37 | 51.01 | 59.03 | 42.91 | 63.63 | 78.79 | 75.24 | 79.92 | 66.95 | 65.70 | 65.58 | 63.62 | 58.01 | 62.52 |
| UniFD [26] $\mathcal{A}$ | 37.01 | 54.91 | 57.11 | 77.13 | 68.64 | 75.41 | 70.71 | 88.32 | 80.71 | 48.81 | 61.70 | 62.00 | 78.25 | 61.39 | 65.86 |
| UniFD [26] $\mathcal{B}$ | 43.21 | 54.85 | 48.93 | 64.09 | 65.95 | 86.45 | 86.62 | 98.97 | 91.66 | 50.28 | 65.87 | 66.25 | 93.90 | 67.60 | 70.33 |
| UniFD [26] $\mathcal{C}$ | 72.38 | 63.50 | 53.84 | 53.39 | 58.83 | 70.90 | 78.07 | 91.84 | 87.39 | 54.51 | 78.84 | 78.88 | 85.03 | 74.03 | 71.53 |
| UniFD [26] $\mathcal{D}$ | 82.89 | 43.82 | 51.41 | 70.09 | 66.11 | 70.32 | 79.99 | 87.83 | 91.46 | 61.48 | 81.65 | 82.24 | 78.37 | 76.10 | 73.13 |
| RINE [80] $\mathcal{A}$ | 79.83 | 69.76 | 59.61 | 70.65 | 85.39 | 79.33 | 70.20 | 96.34 | 94.82 | 56.52 | 80.10 | 78.37 | 95.42 | 90.59 | 79.07 |
| RINE [80] $\mathcal{B}$ | 85.92 | 66.64 | 57.07 | 84.32 | 73.46 | 98.81 | 95.23 | 99.11 | 98.86 | 53.50 | 96.34 | 95.85 | 97.54 | 93.83 | 86.46 |
| RINE [80] $\mathcal{C}$ | 83.46 | 65.94 | 55.71 | 81.80 | 77.42 | 75.17 | 78.74 | 93.39 | 82.23 | 55.85 | 81.36 | 82.46 | 83.74 | 70.60 | 76.28 |
| RINE [80] $\mathcal{D}$ | 80.57 | 71.22 | 61.16 | 72.79 | 80.98 | 76.95 | 83.42 | 97.90 | 95.50 | 65.64 | 87.92 | 81.38 | 86.43 | 75.81 | 79.83 |
| RGB (ours) $\mathcal{A}$ | 62.18 | 65.52 | 67.22 | 76.64 | 93.80 | 92.26 | 95.93 | 99.96 | 99.09 | 67.46 | 96.48 | 95.87 | 99.91 | 95.74 | 86.29 |
| RGB (ours) $\mathcal{B}$ | 56.05 | 71.54 | 62.66 | 87.17 | 97.71 | 99.25 | 96.43 | 99.99 | 99.85 | 52.69 | 96.41 | 95.85 | 99.98 | 97.08 | 86.62 |
| RGB (ours) $\mathcal{C}$ | 54.88 | 67.92 | 59.93 | 81.59 | 95.06 | 96.49 | 96.16 | 99.98 | 99.45 | 62.86 | 96.53 | 96.00 | 99.96 | 96.13 | 85.92 |
| RGB (ours) $\mathcal{D}$ | 57.42 | 75.21 | 63.66 | 85.86 | 98.01 | 98.28 | 96.58 | 99.99 | 99.85 | 62.30 | 97.34 | 96.92 | 99.93 | 96.64 | 87.71 |
| FreqSpec [25] $\mathcal{A}$ | 47.27 | 57.26 | 50.71 | 54.55 | 76.53 | 75.88 | 49.66 | 95.80 | 81.56 | 48.25 | 60.85 | 60.08 | 53.39 | 64.57 | 62.60 |
| FreqSpec [25] $\mathcal{B}$ | 59.35 | 62.55 | 50.55 | 59.72 | 62.28 | 61.78 | 56.61 | 98.94 | 72.15 | 46.66 | 57.50 | 57.02 | 62.67 | 51.56 | 61.38 |
| FreqSpec [25] $\mathcal{C}$ | 73.26 | 54.00 | 53.36 | 71.77 | 52.98 | 71.17 | 60.10 | 92.27 | 87.60 | 74.48 | 82.07 | 81.92 | 68.30 | 65.91 | 70.66 |
| FreqSpec [25] $\mathcal{D}$ | 70.29 | 53.29 | 55.93 | 67.98 | 49.63 | 64.82 | 55.46 | 96.39 | 92.47 | 75.96 | 83.64 | 83.83 | 63.08 | 68.96 | 70.12 |
| FFiT (ours) $\mathcal{A}$ | 67.64 | 57.70 | 56.18 | 63.55 | 83.20 | 79.15 | 42.84 | 95.05 | 86.09 | 63.46 | 73.16 | 72.02 | 54.26 | 67.10 | 68.67 |
| FFiT (ours) $\mathcal{B}$ | 84.97 | 66.91 | 44.77 | 62.15 | 64.69 | 66.48 | 51.54 | 98.90 | 80.46 | 58.59 | 79.33 | 77.75 | 71.32 | 71.60 | 69.96 |
| FFiT (ours) $\mathcal{C}$ | 87.31 | 60.44 | 57.20 | 70.95 | 71.95 | 62.73 | 64.23 | 56.59 | 59.03 | 49.14 | 69.58 | 68.69 | 61.91 | 66.03 | 64.70 |
| FFiT (ours) $\mathcal{D}$ | 80.78 | 52.33 | 57.21 | 71.22 | 71.01 | 75.97 | 71.05 | 77.45 | 78.95 | 51.93 | 74.54 | 73.61 | 64.61 | 67.96 | 69.19 |
| FatFormer [42] $\mathcal{A}$ | 82.08 | 65.88 | 58.25 | 72.11 | 75.76 | 86.87 | 78.30 | 95.13 | 94.23 | 58.19 | 85.90 | 82.34 | 95.77 | 86.93 | 79.84 |
| FatFormer [42] $\mathcal{B}$ | 87.01 | 68.35 | 59.57 | 89.81 | 95.12 | 98.61 | 95.24 | 99.70 | 99.36 | 54.95 | 93.24 | 91.49 | 98.64 | 95.41 | 87.61 |
| FatFormer [42] $\mathcal{C}$ | 88.25 | 70.14 | 54.09 | 76.75 | 80.33 | 73.85 | 85.20 | 90.32 | 88.61 | 56.81 | 80.84 | 79.30 | 81.90 | 69.54 | 76.85 |
| FatFormer [42] $\mathcal{D}$ | 83.77 | 65.57 | 58.13 | 80.85 | 85.56 | 79.21 | 88.68 | 98.54 | 93.57 | 57.81 | 82.90 | 81.39 | 85.72 | 82.40 | 80.29 |
| multimodal (Ours) $\mathcal{A}$ | 74.42 | 64.13 | 60.59 | 65.10 | 89.54 | 91.23 | 72.52 | 99.43 | 96.10 | 61.28 | 88.33 | 87.49 | 96.44 | 88.57 | 81.08 |
| multimodal (Ours) $\mathcal{B}$ | 90.71 | 70.03 | 60.48 | 88.46 | 97.04 | 99.03 | 94.44 | 99.99 | 99.82 | 56.58 | 97.25 | 96.71 | 99.90 | 97.33 | 89.13 |
| multimodal (Ours) $\mathcal{C}$ | 85.95 | 68.59 | 58.01 | 74.68 | 82.12 | 77.60 | 79.60 | 92.83 | 85.54 | 51.81 | 81.63 | 80.70 | 84.98 | 79.60 | 77.40 |
| multimodal (Ours) $\mathcal{D}$ | 84.36 | 69.02 | 59.47 | 76.92 | 84.88 | 89.83 | 84.74 | 97.75 | 95.75 | 51.79 | 85.87 | 85.11 | 89.14 | 81.93 | 81.18 |

## 4.3 Task 2: Cross-time Evaluation on the NeuroRenderedFake Dataset

[24] organized the fake images generated by various diffusion models based on the release dates of their respective generation methods. It was observed that fake detectors trained on earlier-released

images show a decline in performance when tested on datasets composed of later-released images. Conversely, detectors trained on later-released fake images perform well on testing datasets containing earlier-released fake images. This finding highlights a trend: as generative model technologies evolve, the artifacts within fake images become progressively more challenging to identify. Inspired by this observation, we similarly organized neural-rendered fake images according to the release dates of their corresponding neural rendering techniques.

In Fig. 4, we visualize the cross-time performance of NeRF-based methods in the first row and 3DGS-based methods in the second row. The method names along the x-axis indicate that the detector is trained using all available images in the training dataset that are synthesized by neural rendering methods released no later than the corresponding method name. The x-axis is sorted chronologically by release date. The y-axis lists the fake image generation methods used in the test dataset. For instance, in the figure located at the first row and first column, the value in the upper right corner indicates that when the detector [1] is trained on real images and fake images generated by methods released before PyNeRF (i.e., i-ngp, tensorF, nerfacto, and SeaThru) as well as PyNeRF itself, and then tested on the PyNeRF test dataset, the AP is 96.63.

Unlike what was found for diffusion models, our cross-time evaluation of neural rendered fake images does not reveal an evident trend similar to that observed in the study of generative models as discussed in [24]. Further details on the organization of the cross-time training protocol are available in Sec. C of the Appendix.

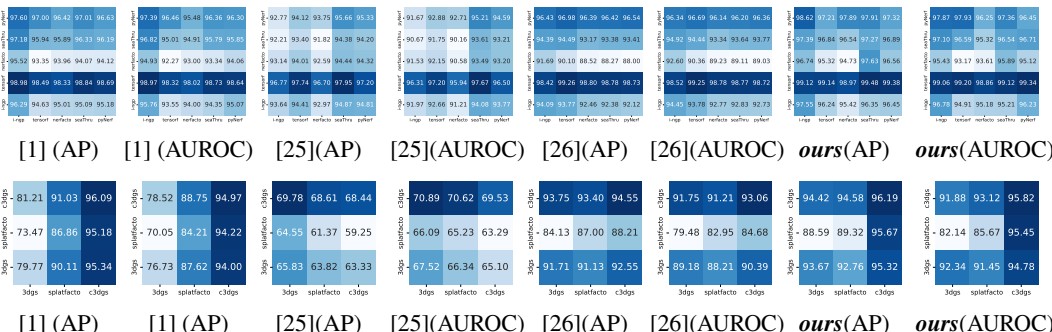

Figure 4: Comparative performances of cross-time fake detection tasks on NeRF and 3DGS data.

## 4.4 Task 3: Degraded NeuroRenderedFake Images Detection

The degradation of fake images produced by generative models has been shown to reduce the effectiveness of fake detection tasks. In this section, we explore how introducing quality degradation to neural rendered fake images impacts detection performance, focusing specifically on the effects of noise. To achieve this, we applied JPEG compression with the quality ratio of 60 and introduced Gaussian blurring with a kernel size of 5. The experimental results evaluated by AUROC are presented in Fig. 5 for testing groups from 1 to 11. Our findings indicate that the degraded quality of neural-rendered fake images negatively affects detection performance. JPEG compression and Gaussian blurring present significant challenges for fake detection tasks, particularly for popular detectors that rely on spectral domain artifacts [74]. This indicates that the block-wise artifacts of JPEG compression and the smoothing effects of Gaussian blurring can pose as challenging noise in the successful detection of neural-rendered fake images.

## 4.5 Assessment of Relative Contribution from Spatial and Spectral domain.

Since the contributions from the spatial and spectral domains are explicitly modeled in our designed multimodal architecture through the weights $w_a$ and $w_b$, with the constraint $w_a + w_b = 1$, we investigate the distribution of $w_b$, which represents the contribution from the spectral domain to the final prediction. We aim to determine the following: Is there a relationship between the $w_b$ values and whether the input image is real or fake? Do the $w_b$ values correlate with the training dataset? Does the distribution of $w_b$ vary across different testing datasets?

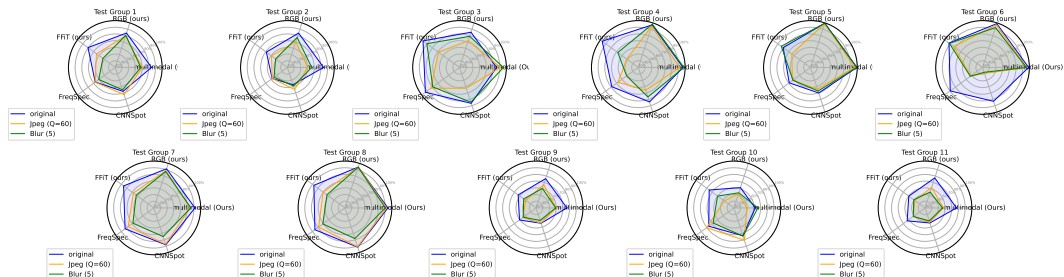

Figure 5: Comparative performances of degraded neural-rendered fake image detection (in %).

In Table 6, we provide the average values of $w_b$ for both real and fake images during inference, for detectors trained on each dataset $\mathcal{A}$, $\mathcal{B}$, $\mathcal{C}$, and $\mathcal{D}$, and tested across groups 1 to 11.

Our observations reveal that when the training dataset is fixed and the detector is applied to the same testing dataset, the contribution from the spectral domain, as indicated by $w_b$, consistently differs between real and fake images if the detector is trained on fake images produced by generative models (i.e., $\mathcal{A}$ and $\mathcal{B}$). Conversely, when our multimodal detector is trained on neural-rendered fake images (i.e., $\mathcal{C}$ and $\mathcal{D}$), there is no significant difference in $w_b$ values between real and fake images. Specifically, for the detector trained on GAN-generated fake images ($\mathcal{A}$), there is a prominent bias towards the spectral domain. This phenomenon likely arises because the discrepancy in spectral energy between real and fake images generated by GANs is more pronounced compared to those produced by diffusion models and neural rendering models (as detailed in the Appendix), leading the multimodal detector to place greater emphasis on the spectral branch during training. Furthermore, for testing groups 9 and 11, where the discrepancy in 1D spectral energy between real and fake images is minimal (details provided in the Appendix), the multimodal detector assigns relatively low weights $w_b$ for incorporating information from the spectral domain. This indicates that the feature fusion module dynamically reduces its dependence on the spectral domain during inference when the discrepancy in energy distribution between real and fake images becomes difficult to distinguish.

Table 6: The average value of $w_b$ for fake images.

|  |  | 1 | 2 | 3 | 4 | 5 | 6 | 7 | 8 | 9 | 10 | 11 |
|---|---|---|---|---|---|---|---|---|---|---|---|---|
| real | $\mathcal{A}$ | 0.622 | 0.623 | 0.598 | 0.598 | 0.592 | 0.593 | 0.617 | 0.614 | 0.591 | 0.623 | 0.580 |
|  | $\mathcal{B}$ | 0.540 | 0.541 | 0.518 | 0.519 | 0.523 | 0.524 | 0.536 | 0.539 | 0.505 | 0.567 | 0.510 |
|  | $\mathcal{C}$ | 0.589 | 0.590 | 0.578 | 0.579 | 0.557 | 0.556 | 0.589 | 0.587 | 0.520 | 0.568 | 0.516 |
|  | $\mathcal{D}$ | 0.606 | 0.607 | 0.552 | 0.553 | 0.521 | 0.520 | 0.606 | 0.604 | 0.483 | 0.617 | 0.478 |
| fake | $\mathcal{A}$ | 0.635 | 0.636 | 0.668 | 0.684 | 0.670 | 0.698 | 0.646 | 0.644 | 0.606 | 0.627 | 0.613 |
|  | $\mathcal{B}$ | 0.557 | 0.572 | 0.613 | 0.655 | 0.607 | 0.626 | 0.584 | 0.616 | 0.506 | 0.530 | 0.496 |
|  | $\mathcal{C}$ | 0.590 | 0.615 | 0.591 | 0.557 | 0.565 | 0.586 | 0.585 | 0.586 | 0.518 | 0.519 | 0.521 |
|  | $\mathcal{D}$ | 0.600 | 0.638 | 0.528 | 0.590 | 0.573 | 0.462 | 0.581 | 0.592 | 0.480 | 0.510 | 0.482 |

## 5 Conclusions and Future Work

This paper introduces *NeuroRenderedFake*, a benchmark dataset specifically designed for detecting fake images generated by neural rendering methods and their derivatives. *NeuroRenderedFake* serves as a million-scale benchmark, addressing the critical shortage of neural rendered fake image datasets for evaluating fake detection methods. We propose cross-domain evaluation tasks for currently popular fake synthesis approaches, in which detectors are trained using fake images with artifacts from generative models and tested on those with artifacts from neural rendering models, and vice versa. This setup enables us to quantify the domain gap between fake images generated by generative models and neural rendering methods. Furthermore, the proposed multimodal architecture achieves SOTA performance, demonstrating that neural-rendered fake images can be effectively distinguished and facilitating further analysis of fake detection tasks on the *NeuroRenderedFake* dataset. An investigation into the contribution of the spectral domain is also provided, offering insights into how multimodal detectors combining spatial and spectral domain features achieve robustness under specific conditions.

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

# SUPPLEMENTARY MATERIAL

By leveraging neural rendering technologies based on NeRF and 3DGS, we create a wide array of realistic 3D scene representations and generate a multitude of synthesized 2D images from different perspectives. Moreover, through the combination of generative models with these advanced neural rendering methods, we generate highly sophisticated but fake images that incorporate combined artifacts. Unlike other existing datasets that largely focus on fake images generated by traditional generative models such as GANs or diffusion models, our *NeuroRenderedFake* dataset significantly extends the boundaries of a much-needed dataset for sophisticated fake image detection. This benchmark consists of over 2 million images, i.e., 512,972 authentic images and 1,653,881 highly sophisticated fake images. Therefore, it can serve as the largest collection of diverse images generated through advanced synthesis and neural rendering techniques.

This work is expected to have a significant positive societal impact, particularly benefiting the forensic community and media outlets. Our method can enhance the accurate and timely identification of real-look-like but fake images that are often found in our mailboxes or social media platforms. The development of accurate techniques to detect these images is crucial for addressing concerns related to security, privacy, and preserving harmony within our community. Importantly, the dataset created in this study does not involve human subjects or their personal data. It was synthesized using publicly accessible sources, which are clearly documented and shared alongside the dataset. Additionally, our synthesized videos and corresponding audio are also generated from publicly available content and are clearly labeled as synthesized (fake). Consequently, there is no risk associated with the release of our datasets, as comprehensive safeguards have been implemented to prevent any potential misuse.

Our supplementary material is organized as follows: **In Section A**, we provide details of the developed multimodal architecture, including its design and training specifications. **In Section B**, we present an analysis of spectral domain imprints, which serves as a supplement to Section 4.5 of the main text. **Section C** describes the detailed training and testing protocols, as well as the construction of the dataset. **Section D** discusses the scenario in which the frequency of fake images is manipulated and examines the corresponding impact on detection performance. **Section E** provides additional visualization samples from the NeuroRenderedFake dataset.

## A    Motivation, Design and Training of Multimodal Architecture

### A.1    Additional Details on Network Training

The spatial branch is initialized with ViT weights pre-trained on the ImageNet dataset, except for the Ada-LoRA modules, which are randomly initialized. The spectral branch is initialized with ViT weights pre-trained by FFiT, which were acquired before, with its Ada-LoRA modules also randomly initialized. Both branches are fine-tuned using a learning rate of $1 \times 10^{-4}$ using the AdamW optimizer and a batch size of 256 on a single H100 GPU for 20 epochs, employing the BCEWithLogits loss. The fine-tuning follows the training protocols of groups $\mathcal{A}$, $\mathcal{B}$, $\mathcal{C}$, and $\mathcal{D}$. After acquiring the fine-tuned spatial and spectral branches, we fix their parameters and fine-tune only the last GMU layer with the FC layer to obtain the optimal parameters for fake image classification. During each training stage, including the fine-tuning of the spatial branch, spectral branch, and GMU module, we use a class-balanced random sampler, following the approach described in [24], to balance the distribution of generated and real images over an epoch. The learning rate for fine-tuning the GMU layer is set to $1 \times 10^{-5}$, with a total of 20 training epochs using the AdamW optimizer.

**Data Augmentation for Training: 1) Spatial Branch:** Initially, all input images were resized to 256x256 pixels. Following this step, a cropping operation was performed where images were randomly cropped to 224x224 pixels. To further diversify the dataset, we applied horizontal and vertical flips with probabilities of 10% each. Additionally, Gaussian blur was applied with a 5% probability to simulate variations in image clarity. Image compression was also introduced with a 10% chance, varying the JPEG quality between 60 and 100 to mimic different levels of image degradation. For normalization, we used the mean and standard deviation values from the ImageNet dataset ([0.485, 0.456, 0.406] and [0.229, 0.224, 0.225], respectively) to standardize the pixel values. **2) Spectral Branch:** We adopt the same data augmentation settings for training the spectral branch as those used for the spatial branch. **3) Multimodal Training:** We adopt the same data augmentation settings for training the spectral branch as those used for the GMU module.

## A.2 The Design of Spectral Branch (FFiT)

We also investigate an unsupervised training approach to enable large models to extract comprehensive features from the Fourier spectrum's magnitude, thereby overcoming the challenges of reconstructing the spectrum due to its centrosymmetric properties. MAE [54] is a classical method to train large neural models in an unsupervised way. However, the centrosymmetric characteristic of the spectrum, wherein the amplitudes at positive frequencies are equivalent to those at the corresponding negative frequencies, thereby exhibiting symmetry about the zero frequency, can introduce adverse effects to the training if we use the same way as MAE to train the Transformer on the spectral domain. In Fig. Aa, a sample of the original magnitude of the spectrum is presented. Fig. Ab displays the mask utilized for patch masking during the inference stage, with the model that is trained using the original MAE-based training strategy. In this mask, white blocks indicate the patches that are to be masked during the patch embedding process, whereas black blocks represent the regions that should remain unmasked. Fig. Ac illustrates the reconstructed magnitude of the spectrum, based on the input from Fig. Aa and the mask shown in Fig. Ab, demonstrating a poor quality of reconstruction. In Fig. Ad, three representative types of regions of Fig. Ac are highlighted, and the following observations can be made:

**case (i):** When both a masked patch and its centrosymmetric counterpart are masked, the pre-trained model is unable to accurately reconstruct either. This indicates a limitation in the model's ability to infer information from the neighboring patches.

**case (ii):** In the case of masked patches for which the corresponding centrosymmetric patches remain unmasked, the pre-trained model demonstrates a capability to reconstruct these patches with high accuracy. This suggests that the model effectively captures and utilizes the centrosymmetric property of the spectrum during training.

**case (iii):** For the unmasked regions, it is evident that the pre-trained model fails to reconstruct them accurately. This finding is contrary to the expected behavior in the spatial domain, where an MAE-trained model typically succeeds in reconstructing unmasked areas.

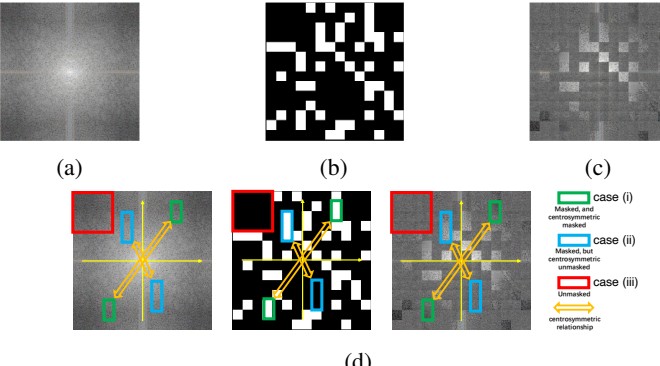

Figure A: Failure in spectral information extraction with the original MAE pre-training. (a) Input spectrum magnitude. (b) Patch embedding mask for inference. (c) Poor-quality reconstruction from (a) and (b). (d) Explanation of reconstructed patches in (c).

## A.3 Balancing the Weights of Various Masking Types

In the original MAE training process, the block-wise reconstruction loss $\mathcal{L}_{B(i,j)}$, which represents the reconstruction error for the $i^{\text{th}}$ row and $j^{\text{th}}$ column patch ($0 \leq i, j \leq N - 1$) between the original input magnitude of spectrum $X$ and the reconstructed $X'$, is calculated as follows:

$$\mathcal{L}_{B(i,j)} = \sum_{m=0}^{W-1}\sum_{n=0}^{W-1}||X(Wi+m,Wj+n)-X'(Wi+m,Wj+n)||^2 \tag{2}$$

where $X$ is divided into $N \times N$ patches ($N$ is an even number) in a Transformer-based architecture. Given that $X$ is of size $224 \times 224$ pixels and each patch is of size $W \times W$ pixels with $W = 16$, we have $N = 224/W = 14$. During the training process, masks are applied to these patches, compelling

the model to reconstruct the patterns within the masked regions, thereby facilitating unsupervised learning.

The total loss function in the original MAE training is computed by summing the reconstruction losses over the masked blocks, i.e., those $B(i, j)$ that are masked. This approach has two key limitations that contribute to the failure to reconstruct the magnitude of the spectrum: 1) Ignorance of unmasked blocks in the loss function: the model is not penalized for any inaccuracies in the unmasked regions, which can lead to a lack of refinement in the overall reconstruction quality. 2) Overlooking centrosymmetric information: a masked block may have an unmasked centrosymmetric counterpart from which information can be easily copied. These limitations highlight the need for a more sophisticated loss function or training strategy that takes into account the unmasked regions and leverages the inherent symmetries within the spectral data to improve the reconstruction performance.

To address the limitations of the original MAE training process, we adopt a modified loss function that incorporates the focal loss mechanism. This approach aims to balance the influence of different masking cases, considering the special properties of the spectral magnitude. The probability of a patch being masked is assumed to be $r$. The loss function, denoted as $\mathcal{L}_{r \neq 0}(X, X'|r)$, is defined as follows:

$$\mathop{\mathcal{L}}_{r \neq 0}(X, X'|r) = -\frac{1}{N^2} \sum_{i=0}^{N-1} \sum_{j=0}^{N-1} \alpha_t (1 - \mathcal{L}_{B(i,j)})^\gamma \log \mathcal{L}_{B(i,j)}, \tag{3}$$

for $B(i, j) \in$ masking case $t$ ($t \in \{1, 2, 3\}$), $\alpha_t$ is used to balance its weight according to the occurring frequency of the specific case.



(a) input       (b) mask       (c) reconstructed       (d) difference

Figure B: During training, we adopt the loss function in Eq. (3) and fix $r$ to 0.3. For inference, the mask ratio of (b) is set to 0.25.

The focusing parameter $\gamma$ is designed to impose a greater penalty on less frequent examples. The balancing factor $\alpha_t$ is used to adjust the contribution of each class based on its effective number of samples. It is calculated as follows:

$$\alpha_t = P_t^{-1} / (P_1^{-1} + P_2^{-1} + P_3^{-1}), \quad t = 1, 2, 3 \tag{4}$$

where $P_1$, $P_2$, and $P_3$ refer to the expected probability for masking cases 1, 2, and 3.

The expected number of pairs of masked blocks for the three different cases, which are denoted as $E_1$, $E_2$, and $E_3$, can be computed as:

$$E_1 = \frac{N^2}{2} \times r^2, E_2 = \frac{N^2}{2} \times 2r \times (1 - r), E_3 = \frac{N^2}{2} \times (1 - r)^2 \tag{5}$$

Thus the probabilities $P_1$, $P_2$, and $P_3$ for three cases are:

$$P_1 = r^2, \quad P_2 = 2r \times (1 - r), \quad P_3 = (1 - r)^2 \tag{6}$$

A reconstructed sample is presented in Fig. B, with the masking ratio during inference set to 0.25. The results indicate that regions corresponding to all three masking cases are reconstructed with high quality.

## A.4 Dynamic Masking Ratio for FFiT Training

Although the reconstructed sample in Fig. B demonstrates high-quality reconstruction with a masking ratio of 0.25 during inference, it can be observed that the global magnitude of the spectrum is not perfectly recovered as shown in Fig. C: when the model is trained with the same settings but evaluated

with a mask ratio of 0, the reconstructed magnitude of the spectrum (Fig. Cb) exhibits inconsistencies between blocks. Additionally, we find that if the mask ratio for inference significantly varies from the mask ratio during training, the performance of spectral reconstruction can be negatively influenced.

This observation inspires us to introduce a dynamic masking mechanism during training, where the mask ratio is randomly varied across different mini-batches. Specifically, we define three levels of masking: heavily masked, slightly masked, and not masked, with corresponding mask ratios $r_1$, $r_2$, and $r_3$ set to 0.3, 0.15, and 0.0, respectively. Within each mini-batch, the mask ratio is consistent (i.e., it is either $r_1$, $r_2$, or $r_3$), but the specific mask ratio used varies between different batches.



| (a) input | (b) reconstructed | (c) difference of (a), (b) |

Figure C: During training, adopt our loss function but set $r$ as a fixed value 0.3. During inference, using mask with ratio of 0.

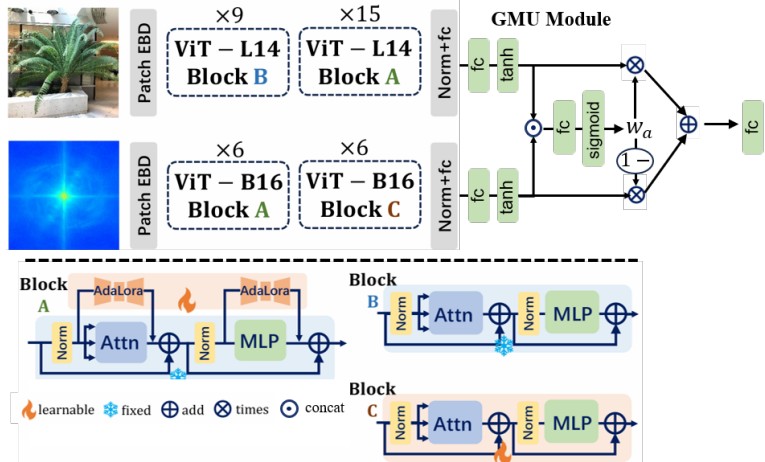

Figure D: Spatial-frequency architecture with the different blocks and distinct fine-tuning strategies across various network stages.

Different from $\mathcal{L}_{r \neq 0}(X, X'|r)$ for $r_1$, $r_2$, when $r_3 = 0$:

$$\mathcal{L}(X, X'|r_3 = 0) = \frac{1}{N^2} \sum_{i=0}^{N-1} \sum_{j=0}^{N-1} \mathcal{L}_{B(i,j)} \tag{7}$$

It was empirically observed that the order of magnitude for the loss function varies with different mask ratios. To mitigate such instability in training caused by significant fluctuations in gradient updates across batches for varying $r$ values, we introduce a scaling factor. Specifically, we compute the expected loss $\mathbb{E}[\mathcal{L}(X, X'|r)]$ for each $r$, and then scale the individual losses $\mathcal{L}(X, X'|r_1)$, $\mathcal{L}(X, X'|r_2)$, and $\mathcal{L}(X, X'|r_3)$ by the reciprocal of their respective expectations. This normalization ensures a more consistent gradient descent process, thereby enhancing the stability of the neural network's training across different mask ratio configurations.

To compute the expectation of the reconstruction loss, we assume that $\mathcal{L}_{B(i,j)}$ follows a $\chi$ distribution and is independent of the scenario type $t$. A detailed derivation proving that $\mathcal{L}_{B(i,j)}$ conforms to a $\chi$ distribution is provided in the Supp. A. Our goal is to determine $\mathbb{E}[\mathcal{L}(X, X'|r_k)]$ for $k = 1, 2, 3$, where $\mathbb{E}[\cdot]$ represents the expectation over the specified distribution.

For $k = 1$ and $k = 2$, the $r_k \neq 0$, we acquire:

$$\mathbb{E}(\mathcal{L}(X, X'|r_k)) = \sum_{t=1}^{3} P_t \mathbb{E}[\mathcal{L}(X, X'|r_k)|t] \tag{8}$$

For each $t$, the $\mathbb{E}[\mathcal{L}(X, X'|r_k)|t]$ is equal to:

$$-\frac{1}{N^2}\sum_{i=0}^{N-1}\sum_{j=0}^{N-1}\alpha_t\mathbb{E}[(1-\mathcal{L}_{B(i,j)})^\gamma\log\mathcal{L}_{B(i,j)}] \tag{9}$$

$$=-\alpha_t\mathbb{E}[(1-\mathcal{L}_B)^\gamma\log\mathcal{L}_B]$$

Therefore, for $k=1$ and $k=2$,

$$\mathbb{E}(\mathcal{L}(X, X'|r_k)) = -\left(\sum_{t=1}^{3}P_t\alpha_t\right)\mathbb{E}[(1-\mathcal{L}_B)^\gamma\log\mathcal{L}_B] \tag{10}$$

$$=-\frac{3r_k^2(1-r_k)^2}{3r_k^2-3r_k+2}\mathbb{E}[(1-\mathcal{L}_B)^\gamma\log\mathcal{L}_B]$$

where $\mathbb{E}[(1-\mathcal{L}_B)^\gamma\log\mathcal{L}_B]$ is equal to:

$$\int_0^\infty (1-x)^\gamma\log x\cdot\frac{1}{2}e^{\frac{-(x+\lambda)}{2}}\left(\frac{x}{\lambda}\right)^{\frac{k}{4}-\frac{1}{2}}I_{k/2-1}(\sqrt{\lambda x})\mathrm{d}x \tag{11}$$

where $I_\nu(z)$ is the modified Bessel function of the first kind of order $\nu$. Detailed steps are provided in the Supp. A.

For $k=3$, we can easily get:

$$\mathbb{E}[\mathcal{L}(X, X'|r_3)] = \mathbb{E}\left[\frac{1}{N^2}\sum_{i=0}^{N-1}\sum_{j=0}^{N-1}\mathcal{L}_{B(i,j)}\right] = \mathbb{E}[\mathcal{L}_B] \tag{12}$$

Therefore, for mask ratio of to 0.3, 0.15, 0, we have the scaled loss function $\mathcal{L}_k$ as:

$$\mathcal{L}(X, X'|r_k)/\mathbb{E}(\mathcal{L}(X, X'|r_k)) \tag{13}$$

We empirically observe that it is necessary to introduce dynamic masking ratios to capture the global reconstruction during pre-training of FFiT. After dynamically setting $r$ to $r_1, r_2, r_3 = 0.3, 0.15, 0$ respectively, the global reconstruction is perfect which can be observed from Fig. E.



| (a) input | (b) mask for (c) | (c) from (a), (b) | (d) from (a) only |

Figure E: Dynamically set $r = 0.3, 0.15, 0$. (a) original magnitude, (b) mask with ratio of 0.25, (c) the reconstructed magnitude using (b) mask, (d) the reconstructed magnitude without mask.

During the pre-training of FFiT, we empirically set the focusing parameter $\gamma$ in the developed loss function to 2. The learning rate is set to $1 \times 10^{-4}$ with a batch size of 256 on a single H100 GPU. We employ early stopping, terminating the training when the reconstruction loss stagnates and does not improve for 5 consecutive epochs.

### A.5 Squared Euclidean Distance between Two Normally Distributed Vectors

In the main text section detailing the developed loss function for the frequency branch, we assume that the patch $\mathbf{X}$ representing the predicted magnitude of the frequency follows a normal distribution $\mathbf{X} \sim N(\boldsymbol{\mu}_1, \Sigma_1)$, while the patch $\mathbf{Y}$ representing the ground truth magnitude of the frequency follows $\mathbf{Y} \sim N(\boldsymbol{\mu}_2, \Sigma_2)$. Here, $\boldsymbol{\mu}_1$ and $\boldsymbol{\mu}_2$ denote the mean vectors, and $\Sigma_1$ and $\Sigma_2$ represent the corresponding covariance matrices.

When computing the squared Euclidean distance between these vectors, we are essentially computing $\mathcal{L}_B = (\mathbf{X} - \mathbf{Y})^\top(\mathbf{X} - \mathbf{Y})$. Letting $\mathbf{Z} = \mathbf{X} - \mathbf{Y}$, then $\mathbf{Z}$ is also a multivariate normal random vector with mean $\boldsymbol{\mu}_Z = \boldsymbol{\mu}_1 - \boldsymbol{\mu}_2$ and covariance matrix $\Sigma_Z = \Sigma_1 + \Sigma_2$ (assuming independence between $\mathbf{X}$ and $\mathbf{Y}$).

The distribution of $\mathbf{Z}^\top \mathbf{Z}$ follows a generalized chi-squared distribution. Specifically, if $\boldsymbol{\mu}_1 = \boldsymbol{\mu}_2$ and $\Sigma_1 = \Sigma_2 = I$, where $I$ is the identity matrix, then $\mathbf{Z}^\top \mathbf{Z}$ would follow a standard chi-squared distribution with $d$ degrees of freedom ($d$ being the dimension of $\mathbf{X}$ and $\mathbf{Y}$). However, when $\boldsymbol{\mu}_1 \neq \boldsymbol{\mu}_2$ or $\Sigma_1 \neq \Sigma_2$, the distribution of $\mathbf{Z}^\top \mathbf{Z}$ is a noncentral chi-squared distribution, which is described as follows.

**Noncentral Chi-Squared Distribution**: For $\mathbf{Z}^\top \mathbf{Z}$, the degrees of freedom $k$ equals the dimension of $\mathbf{Z}$, and the noncentrality parameter $\lambda$ is given by:

$$\lambda = \boldsymbol{\mu}_Z^\top \Sigma_Z^{-1} \boldsymbol{\mu}_Z$$

Thus, the distribution can be written as:

$$\mathbf{Z}^\top \mathbf{Z} \sim \chi^2(k, \lambda)$$

The probability density function (PDF) of a noncentral chi-squared distribution with $k$ degrees of freedom and noncentrality parameter $\lambda$ is given by:

$$f(x; k, \lambda) = \frac{1}{2} e^{-(x+\lambda)/2} \left(\frac{x}{\lambda}\right)^{(k/4 - 1/2)} I_{k/2-1}(\sqrt{\lambda x})$$

where $I_\nu(z)$ is the modified Bessel function of the first kind of order $\nu$.

Let $\mathcal{L}_B$ be a variable that follows the noncentral chi-squared distribution defined above, therefore, $\mathbb{E}[(1 - \mathcal{L}_B)^\gamma \log \mathcal{L}_B]$ can be computed as:

$$\int_0^\infty (1 - x)^\gamma \log(x) f(x; k, \lambda) \, \mathrm{d}x$$

# B  The Analysis of Spectral Domain Imprints

## B.1  Discrepancy of Magnitude of Spectrum between Real and Fake for NeuroRenderedFake

We visualize the averaged magnitude of the spectrum for real images and various fake images generated by exclusive neural rendering methods from the NeuroRenderedFake database, in Fig. F. Additionally, we illustrate the differences between the magnitude of the spectrum produced by these neural rendering methods and that of real images.

We visualize the 1D spectral energy distribution of real and fake images generated by exclusive neural rendering methods from the NeuralRenderedFake database in Fig. G. The azimuthal integrated 1D spectral energy $AI(\omega_k)$ for the input magnitude of spectrum $\mathcal{F}(I)$ is computed following the same definition in [41], and is given in Eq. (14):

$$AI(\omega_k) = \int_0^{2\pi} \|\mathcal{F}(I)(\omega_k \cdot \cos(\phi), \omega_k \cdot \sin(\phi))\|^2 \, d\phi \quad \text{for } k = 0, \ldots, M/2 - 1, \qquad (14)$$

| i-ngp | tensorf | nerfacto | seathru | pynerf | 3dgs | splatfacto | c3dgs |
|-------|---------|----------|---------|--------|------|------------|-------|

| real | Diff | Diff | Diff | Diff | Diff | Diff | Diff |
|------|------|------|------|------|------|------|------|

Figure F: The 2D spectral energy discrepancy among the exclusive neural rendered fake images.

We visualize the averaged magnitude of the spectrum for real images and various fake images from the test group 3-11 of the NeuroRenderedFake database, in Fig. H. In Fig. H, the first row shows

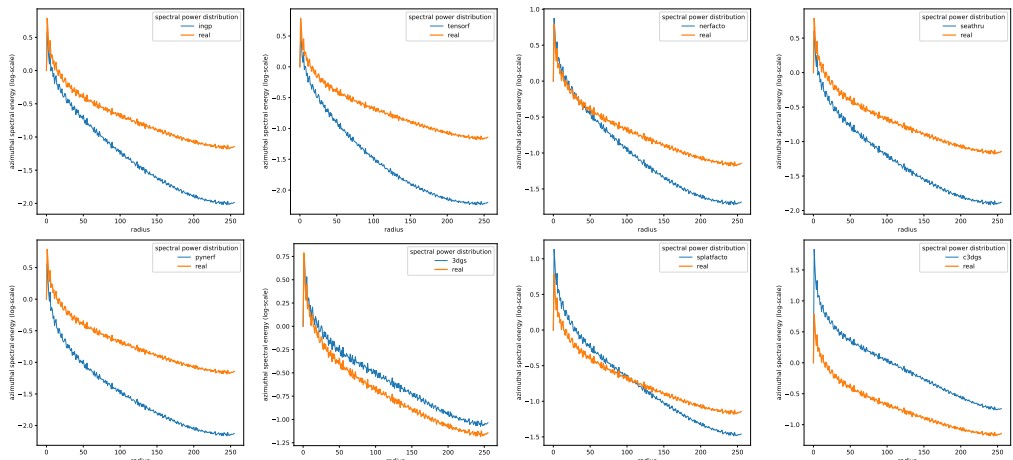

Figure G: The 1D spectral energy discrepancy among the exclusive neural rendered fake images.

the magnitude of the spectrum of real images, the second row shows that of fake images, and the third row displays the difference between the spectra of real and fake images. We also visualize their corresponding 1D spectral energy distribution in Fig. I.

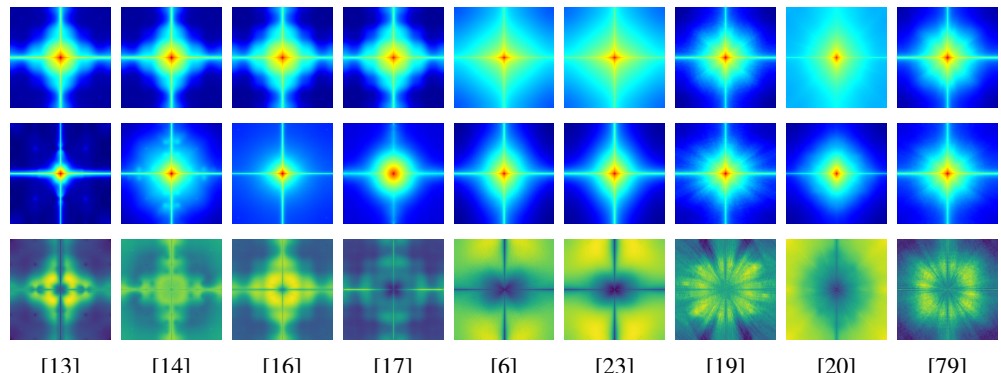

| [13] | [14] | [16] | [17] | [6] | [23] | [19] | [20] | [79] |

Figure H: The 2D spectral energy discrepancy of the test group 3-11 of NeuroRenderedFake.

We define the summation of the 1D spectral energy discrepancy in Eq. (15) and give the corresponding discrepancy values for test group 3-11 in Table I.

$$discrepancy = \int_0^{M/2-1} |AI(\omega_k)_{real} - AI(\omega_k)_{fake}| \mathrm{d}k \tag{15}$$

Table I: Summation of 1D spectral energy discrepancies of test group 3-11 in NeuralRenderedFake

| pix2nerf | sketchfacenerf | dreamfusion | gsgen | in2n | igs2gs | genefacepp | splattingavatar | gaussiantalker |
|----------|----------------|-------------|-------|------|--------|------------|-----------------|----------------|
| [13] | [14] | [16] | [17] | [6] | [23] | [19] | [20] | [79] |
| 47.794 | 25.681 | 60.008 | 47.805 | 71.531 | 75.195 | 3.529 | 19.976 | 3.812 |

## B.2 The Contribution from Spectral Domain Imprints for Multimodal Detector

The contributions from spectral domain branch of the developed multimodal architecture is explicited represented by the learnable parameter $w_b$, and the distribution of $w_b$ varies between real and fake images, varies between different training groups among $\mathcal{A}$, $\mathcal{B}$, $\mathcal{C}$ and $\mathcal{D}$, and also varies between different testing groups ranging from 1 to 11. Therefore, we display the violin plot to represent the distribution for the $w_b$ in Fig. J.

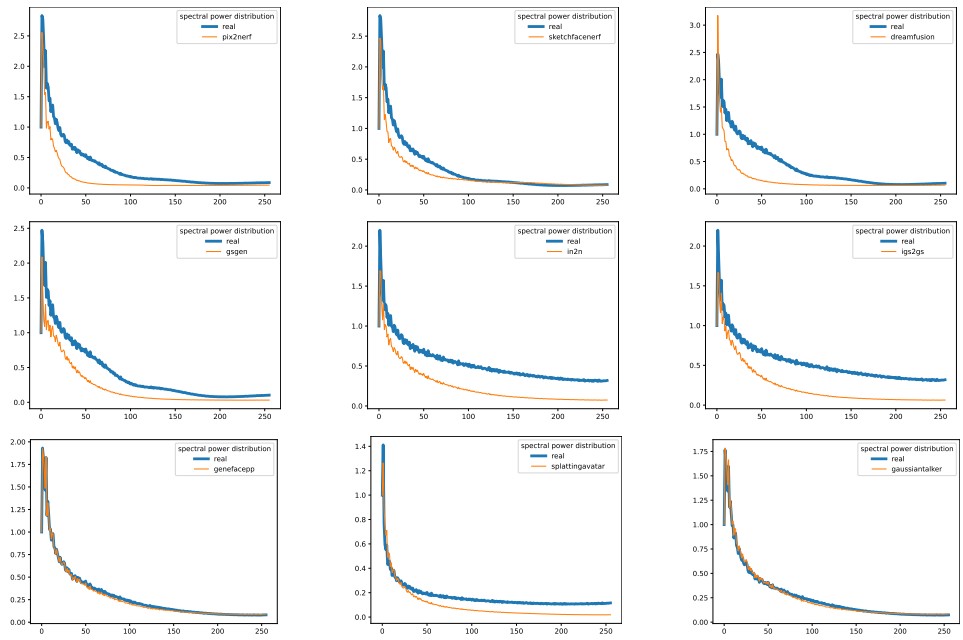

Figure I: The 1D spectral energy discrepancy for the test group 3-11 images in NeuroRenderedFake.

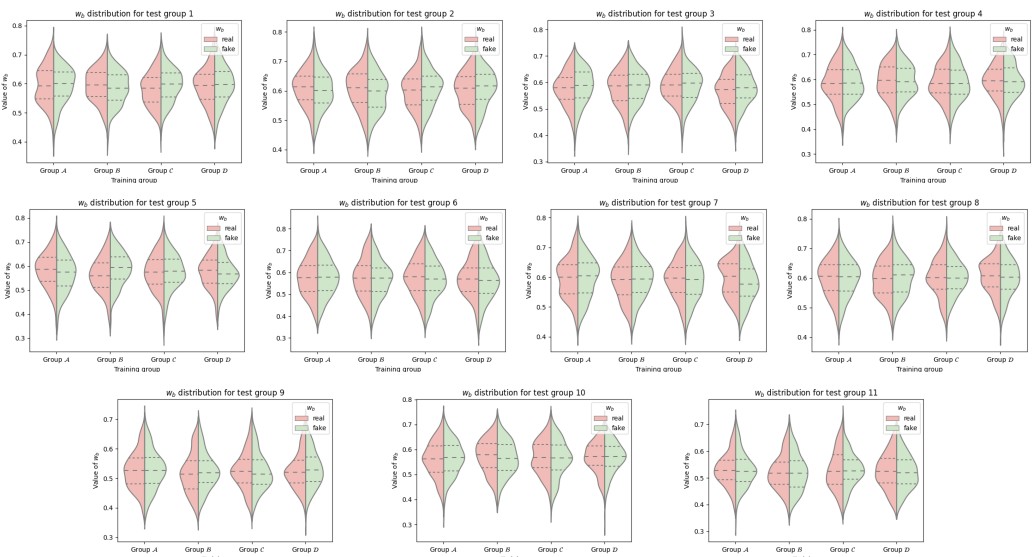

Figure J: The distribution of $w_b$ for different training and testing groups.

# C Details on the Dataset and Protocols

## C.1 Dataset Split Protocol for Training the Detectors

$\mathcal{A}$: For the real images, we randomly select 20,000 images from each of the category in afhq, celebahq, and lsun of ArtiFact [15] database, respectively, and collect all the 4,318 images from landscape class and all the 1,336 images from metfaces class. Therefore we acquire a total of 65,654 real images. For GAN-generated fake images, we collect 10k, 10k, 7k, 10k, 15k, and 15k images from the categories of BigGAN, Gans-former, GauGAN, ProjectedGAN, StyleGAN3, and Taming-Transformer, respectively.

$\mathcal{B}$: The real images are the same as $\mathcal{A}$ while a total of 66,896 fake images generated by 6 DMs are selected. Exactly, we collect 10k, 896, 20k, 6k, 20k, and 10k images from the categories of Glide, DDPM, Latent Diffusion, Palette, Stable Diffusion, and VQ Diffusion, respectively.

$\mathcal{C}$: For the real class, we use all the 69,377 real images in **A**$\sim$**J**. For the rendered class, we collect 12,000 images in **A**$\sim$**J** for method **I**, **II**, **III**, **IV**, **V**, respectively. Therefore, we acquire a total of 60,000 rendered images.

$\mathcal{D}$: For the real class, we use all the 69,377 real images in **A**$\sim$**J**. For the rendered class, we use all the 40,734 splatfacto-rendered images in **A**$\sim$**J** and all the 10,785 C3dgs-rendered images in **G**. Therefore, we acquire a total of 51,519 rendered images.

## C.2 Dataset Split protocol for Performance Evaluation

To evaluate the performance for group 1 and group 2, we select the scenes and the corresponding 2D images that never occur in the training dataset. The details are provided as follows:

Group 1 (**I** $\sim$ **V**): For the real class, we collect all the real images from **K**, **L**, **M**, **N**, **O**. For the fake class, we collect all the images from **K**, **L**, **M**, **N**, **O** rendered by the method **I**, **II**, **III**, **IV**, **V**, respectively.

Group 2 (**V** $\sim$ **VIII**): For the real class, we collect all the real images from **K**, **L**, **M**, **N**, **O**. For the fake class, we collect all the images from **K**, **L**, **M**, **N**, **O** rendered by the method **VI**, **VII**, **VIII**, respectively.

Besides evaluating the performance of unseen fake images generated by NeRF or 3DGS, further consideration is given to scenarios where traditional generative methods, such as GANs and DMs, are combined with neural rendering techniques in groups 3, 4, 5, and 6. Additionally, the use of editable neural rendering methods is explored. In groups 7 and 8, two representative methods capable of editing 3D scenes within their 3D representations are selected. A series of prompts for 3D editing are used, and these edited 3D scenes are then projected into 2D to acquire the fake images. Another important application of neural rendering technologies, digital human (avatar) generation, is also considered. In groups 9 and 10, these technologies are used to generate images of avatars, including both heads and full bodies. In group 11, frames sampled from Sora-generated videos, which exhibit realistic 3D representations within the video, are collected.

**Pix2NeRF[13]:** GAN+Nerf Image-to-image generation. For real class, We use all the 70,000 images in ffhq class of ArtiFact [15] database. For fake class, we render 96,000 ($1,000 \times 96$) images, where we reconstruct 1000 identities and render 96 images from different views for each identity.

**SketchFaceNeRF[14]:** GAN+Nerf Image-to-image generation. For real class, We use all the 70,000 images in ffhq class of ArtiFact [15] database. For fake class, we render 90,000 ($60 \times 60 \times 25$) images, where we use 60 sketches to style-transfer 60 identities and render 25 images from different views for each style-transferred head.

**DreamFusion[16]:** DM+Nerf text-to-image generation. For real class, we randomly collect 10,000 images in imagenet class of ArtiFact [15] database. For the fake class, we render 10,600 ($106 \times 100$) images, where we use 106 prompts for generation and render 100 images from different views for each generated 3D scene.

**GSGEN[17]:** DM+3DGS text-to-image generation. For real class, we randomly collect 10,000 images in imagenet class of ArtiFact [15] database. For the fake class, we render 9,540 ($106 \times 90$)

images, where we use 106 prompts for generation and render 90 images from different views for each generated 3D scene.

**Instruct-N2N[6]:** DM+Nerf image-to-image editing. For real class, we acquire all the 3,174 real images which are used to successfully train the nerfacto (**III**) of dataset **K**, **L**, **M**, **N**, **O**. For the fake class, we generate 40,559 edited images from the nerfacto-generated 3D scenes. The details of this Instruct-N2N dataset can be found in Table II.

**Instruct-GS2GS[23]:** DM+3DGS image-to-image editing. For real class, we acquire all the 3,174 real images which are used to successfully train the splatfacto (**VII**) of dataset **K**, **L**, **M**, **N**, **O**. For the fake class, we generate 40,559 edited images from the splatfacto-generated 3D scenes. The details of this Instruct-GS2GS dataset can be found in Table II.

Table II: Additional details on our instruct-N2N and instruct-GS2GS dataset (generated scenes/rendered images)

| | Prompts | instruct-N2N | | | | | instruct-GS2GS | | | | |
| | | **K** [3] | **L** [4] | **M** [6, 5] | **N** [7] | **O** [8] | **K** [3] | **L** [4] | **M** [6, 5] | **N** [7] | **O** [8] |
|---|---|---|---|---|---|---|---|---|---|---|---|
| prompts for human | Indian attire | ✗ | ✗ | 4/353 | ✗ | ✗ | ✗ | ✗ | 4/353 | ✗ | ✗ |
| | Mustache | ✗ | ✗ | 4/353 | ✗ | ✗ | ✗ | ✗ | 4/353 | ✗ | ✗ |
| | Bronze statue | ✗ | ✗ | 4/353 | ✗ | ✗ | ✗ | ✗ | 4/353 | ✗ | ✗ |
| | Joker makeup | ✗ | ✗ | 4/353 | ✗ | ✗ | ✗ | ✗ | 4/353 | ✗ | ✗ |
| | Gothic makeup | ✗ | ✗ | 4/353 | ✗ | ✗ | ✗ | ✗ | 4/353 | ✗ | ✗ |
| | Anime eyes | ✗ | ✗ | 4/353 | ✗ | ✗ | ✗ | ✗ | 4/353 | ✗ | ✗ |
| | Vintage sepia tone | ✗ | ✗ | 4/353 | ✗ | ✗ | ✗ | ✗ | 4/353 | ✗ | ✗ |
| | Neon lights | ✗ | ✗ | 4/353 | ✗ | ✗ | ✗ | ✗ | 4/353 | ✗ | ✗ |
| | Cyberpunk style | ✗ | ✗ | 4/353 | ✗ | ✗ | ✗ | ✗ | 4/353 | ✗ | ✗ |
| | Renaissance painting | ✗ | ✗ | 4/353 | ✗ | ✗ | ✗ | ✗ | 4/353 | ✗ | ✗ |
| | Pop art | ✗ | ✗ | 4/353 | ✗ | ✗ | ✗ | ✗ | 4/353 | ✗ | ✗ |
| | Tribal face paint | ✗ | ✗ | 4/353 | ✗ | ✗ | ✗ | ✗ | 4/353 | ✗ | ✗ |
| | Alien features | ✗ | ✗ | 4/353 | ✗ | ✗ | ✗ | ✗ | 4/353 | ✗ | ✗ |
| | Pixel art | ✗ | ✗ | 4/353 | ✗ | ✗ | ✗ | ✗ | 4/353 | ✗ | ✗ |
| | Watercolor effect | ✗ | ✗ | 4/353 | ✗ | ✗ | ✗ | ✗ | 4/353 | ✗ | ✗ |
| | Sketch drawing | ✗ | ✗ | 4/353 | ✗ | ✗ | ✗ | ✗ | 4/353 | ✗ | ✗ |
| | Surreal distortion | ✗ | ✗ | 4/353 | ✗ | ✗ | ✗ | ✗ | 4/353 | ✗ | ✗ |
| | Film noir | ✗ | ✗ | 4/353 | ✗ | ✗ | ✗ | ✗ | 4/353 | ✗ | ✗ |
| | Glitch art | ✗ | ✗ | 4/353 | ✗ | ✗ | ✗ | ✗ | 4/353 | ✗ | ✗ |
| prompts for nature | Snowy landscape | 9/1,940 | 8/305 | ✗ | 2/488 | 4/88 | 9/1,940 | 8/305 | ✗ | 2/488 | 4/88 |
| | summer style | 9/1,940 | 8/305 | ✗ | 2/488 | 4/88 | 9/1,940 | 8/305 | ✗ | 2/488 | 4/88 |
| | Autumn foliage | 9/1,940 | 8/305 | ✗ | 2/488 | 4/88 | 9/1,940 | 8/305 | ✗ | 2/488 | 4/88 |
| | spring style | 9/1,940 | 8/305 | ✗ | 2/488 | 4/88 | 9/1,940 | 8/305 | ✗ | 2/488 | 4/88 |
| | Tropical paradise | 9/1,940 | 8/305 | ✗ | 2/488 | 4/88 | 9/1,940 | 8/305 | ✗ | 2/488 | 4/88 |
| | Ancient style | 9/1,940 | 8/305 | ✗ | 2/488 | 4/88 | 9/1,940 | 8/305 | ✗ | 2/488 | 4/88 |
| | High brightness | 9/1,940 | 8/305 | ✗ | 2/488 | 4/88 | 9/1,940 | 8/305 | ✗ | 2/488 | 4/88 |
| | Halloween theme | 9/1,940 | 8/305 | ✗ | 2/488 | 4/88 | 9/1,940 | 8/305 | ✗ | 2/488 | 4/88 |
| | Cosmic style | 9/1,940 | 8/305 | ✗ | 2/488 | 4/88 | 9/1,940 | 8/305 | ✗ | 2/488 | 4/88 |
| | Industrial chic | 9/1,940 | 8/305 | ✗ | 2/488 | 4/88 | 9/1,940 | 8/305 | ✗ | 2/488 | 4/88 |
| | cyberpunk | 9/1,940 | 8/305 | ✗ | 2/488 | 4/88 | 9/1,940 | 8/305 | ✗ | 2/488 | 4/88 |
| | Baroque inspiration | 9/1,940 | 8/305 | ✗ | 2/488 | 4/88 | 9/1,940 | 8/305 | ✗ | 2/488 | 4/88 |

**GeneFace++[19]:** Speech-driven Avatar NeRF. We have 29 videos of different identities (Some source videos are the examples offered by GeneFace++, and the rest of the source videos are publicly available from YouTube, such as subject1, subject2, subject3, subject4, subject5, subject6, subject7, subject8, subject9, subject10, subject11, subject12, etc.) to train the 3D representation of the speaker's head. The original speaker's voice of different identities includes various languages, and such a multi-lingual property leads to the rich diversity of the dataset. We use two different ways to generate the fake speech video: 1). 14 identities to speak the contents from the other 13 identities by inputting the extracted audio, and therefore generate 182 (14 ×13) fake videos. 2) 15 identities to speak a predefined context by using 17 multi-langual, and therefore generate 255 (15x17) fake videos. For the real class, We collect all frames from each real video and generate 88,737 images. For the fake class, we collect all frames from each fake video and generate 452,653 images.

It can be observed from the performance evaluation on this dataset in Section 4 of our paper that the accuracy for fake detection on this dataset is quite low. Therefore, it's a quite challenging dataset that requires further work to address open problem on fake detection and is made available in the public domain to further advance much-needed research in this area.

**SplattingAvatar[20]:** Avatar 3DGS of head/body synthesis. For the real class, we use all the 33,728 images of 14 identities (10 identities are head and 4 identities are full body) provided by [20]. For the fake class, we generate 33,715 rendered images for 14 identities.

**GaussianTalker [79]:** Speech-driven Avatar 3DGS. We extend the real video dataset with one more identity and employ the same videos and same rule 1) and 2) to generate fake speech videos as those used to create GenFace++ fake videos. But only replace the Genface++ method to the Gaussiantalker that uses the 3D Gaussian Splatting. For the real class, we extract all frames from each real video and generate 94,810 images. Conversely, for the fake class, we extract all frames from each fake video and generate 411,401 images.

**SORA[22] frames:** For real class, we randomly acquire 60,000 images in coco class of ArtiFact [15] database. For the fake class, we collected 94 publicly released videos generated by SORA and randomly cropped a total of 60,531 images in the size of $512 \times 512$ pixels from the frames of these SORA-generated videos.

**Liveportrait [77]:** For the real class, we acquire a total of 59,260 frames from 28 videos. For the fake class, we acquire 88,466 frames from 234 generated fake videos. All-to-all matching protocol is adopted for evaluation.

**Runway [21]:** We randomly select 156 real videos from InternVid-10m, and 44 real videos from Youtube-8m dataset. For generating the fake video by the Runway Gen4-Turbo model, we randomly select one image from by randomly select other 161 videos from IngerVid-10m and 35 videos from Yotube-8m. We follow two method to generate the fake video: 1) For the image from the Youtube-8m, we use the image-to-video method, 2) For the image from the InternVid-10m, we use the text&image-to-video to generate the fake video, and the text is the caption of the video from the InternVid-10m. Because the generated fake video by Runway is 24 fps per second, and the total length is 5 seconds with about 121 frames. For the real class, we extract 121 frames from each real video and generate 23,334 images. Conversely, for the fake class, we extract all frames from each fake video and generate 23,353 images.

**106 Prompts Used for Generation of Stable-DreamFusion and GSGEN:** Acropolis of Athens, Desert cactus, Jamaican jerk chicken, Red panda, African elephant, Dolphin, Japanese ramen, Redwood forest, African lion, Dutch pancakes, Japanese sushi, Rhinoceros, Alpine meadow, Egyptian koshari, King cobra, Rose garden, Amazon jungle, Eiffel Tower, Koala bear, Russian borscht, American burger, Emperor penguin, Korean barbecue, Sagrada Familia, Angkor Wat, Ethiopian injera, Korean bibimbap, Saint Basil Cathedral, Arctic wolf, French bakery, Lavender fields, Siberian tiger, Argentine steak, French crepes, Lebanese falafel, Snow leopard, Australian steak, German sausages, Machu Picchu, Spanish tapas, Bald eagle, Giant panda, Malaysian satay, Statue of Liberty, Bamboo forest, Giraffe, Maple tree, Sunflower field, Belgian waffles, Golden Gate Bridge, Mexican churros, Swedish meatballs, Bengal tiger, Gray wolf, Mexican tacos, Swiss chocolate, Blue whale, Great Barrier Reef, Moroccan couscous, Sydney Opera House, Bonsai tree, Great Wall of China, Neuschwanstein Castle, Taj Mahal, Brandenburg Gate, Great white shark, Notre Dame Cathedral, Thai curry, Brazilian barbecue, Greek salad, Oak tree, Thai mango sticky rice, British fish and chips, Grizzly bear, Orca whale, Tower Bridge, Burj Khalifa, Hagia Sophia, Orchid garden, Tropical rainforest, Canadian poutine, Hawaiian poke bowl, Palm tree, Tulip garden, Cheetah, Hippopotamus, Peruvian ceviche, Turkish kebab, Cherry blossom tree, Indian curry, Petra Jordan, Venus flytrap, Chimpanzee, Indian samosas, Pine forest, Water lily pond, Chinese dumplings, Irish stew, Polar bear, Westminster Abbey, Colosseum, Italian gelato, Red fox, Coral reef, Italian pasta, Red kangaroo.

### C.3 Data Split Protocol for Cross-time Performance Evaluation

We especially define two self-evaluation protocols (PT-NN and PT-GG) to observe the performance of detectors which are trained using the past samples and evaluated for their future performance: i) PT-NN: train on NeRF-rendered images and test on unseen NeRF-rendered images, ii) PT-GG: train on 3DGS-rendered images and test on unseen 3DGS-rendered images. For PT-NN and PT-GG, the list of such additional evaluations is summarized in the Table III.

**PT-NN:** To train the detectors, we select the images from the methods of real, **I**, **II**, **III**, **IV**, and **V**, excluding those from categories designated for evaluation, which are summarized in the list of evaluation. To evaluate the performance of the detectors, we utilize all images from the methods of real, **I**, **II**, **III**, **IV**, and **V** located within the categories in the following list that is specified

Table III: Dataset split protocols for cross-times evaluation (PT-NN and PT-GG)

| | method | releasing date | training | | evaluation | |
|---|---|---|---|---|---|---|
| | | | scenes | images | scenes | images |
| **PT-NN** | real | / | 115 | 62,286 | 24 | 10,817 |
| | **I** [9] | 2022 Jan | 91 | 42,110 | 24 | 10,817 |
| | **II** [10] | 2022 Mar | 85 | 38,658 | 24 | 10,817 |
| | **III** [2] | 2023 Feb | 98 | 59,256 | 24 | 10,817 |
| | **IV** [8] | 2023 Apr | 73 | 23,740 | 24 | 10,817 |
| | **V** [11] | 2023 Nov | 60 | 16,640 | 24 | 10,817 |
| **PT-GG** | real | / | 20 | 1,786 | 9 | 1,940 |
| | **VI** [7] | 2023 Aug | 19 | 1,721 | 9 | 1,940 |
| | **VII** [2] | 2023 Sep | 18 | 1,234 | 9 | 1,940 |
| | **VIII** [12] | 2024 Feb | 19 | 1,721 | 9 | 1,940 |

for evaluation: nerfstudio/{bww-entrance, campanile, desolation, Egypt, kitchen, library, person, redwoods2, storefront, stump, vegetation}, record3d/bear, head/face, eyefultower/{office1b, office-view2, riverview}, mip360/{bicycle, bonsai, counter, flowers, kitchen, room, stump, treehill}.

**PT-GG** To train the detectors, we select the 1,786, 1,721, 1,234, 1,721 images from the **L**, **M**, **N**, **O** datasets for the methods of real, **VI**, **VII**, and **VIII**. To evaluate the performance of the detectors, we use the images from the **K** dataset for the methods of real, **VI**, **VII**, and **VIII**.

We summarize the evaluation performance on PT-NN and PT-GG for cross-time evaluation of NeRF and 3DGS, respectively. We sort the neural rendering methods according to the release date, for example, "≤ nerfacto" means we use the fake images generated by i-ngp, tensorf and nerfacto for training. The results in [24] indicate that the testing performance of the detector on the recently released generative models can benefit from more exposure to fake images generated by newly developed methods during training. However, such a trend is not observed for NeRF and 3DGS-generated fake image detection.

## D   Evaluation of Influence from Compromised Fake Images for Detection

One key challenge in the accurate detection of fake images is related to their modification or spectral alignment after synthesis, which makes it challenging to detect them with existing methods. As pointed out in [74], spectral-based fake detectors for generative models heavily rely on the differences in the one-dimensional spectral energy distribution between real and fake images. When the spectrum magnitude of fake images is compromised by replacing their 1D spectral energy distribution with the most similar one from a real image, the detection performance of spectral-based detectors can drop to nearly 50–50. In contrast, spatial-based detectors remain robust to such spectral domain manipulations. In this paper, we aim to investigate whether a similar phenomenon exists in neural rendering-based fake images—specifically, whether spatial-based detectors remain robust to fake images whose frequency content has been compromised, while spectral-based detectors are not.

In Fig. K, we visualize the evaluation of fake detection performance on test groups 1-11, where the fake images have been post-processed using methods developed in [74] to make the 1D spectral energy of the fake images resemble that of real images. Our findings indicate that this phenomenon is also present in fake images generated through neural rendering. Specifically, while spatial-based detectors maintain robustness with only a limited drop in performance, the performance of spectral-based detectors significantly decreases when the frequency of fake images is compromised.

## E   Additional Details on Performance Evaluation

We select several representative methods for comparison. We don't compare with NPR [43] since reference [44] provides a fair comparison of UniFD [26] with NPR [43] and underlines the superiority of UniFD [26] over NPR [43]. We don't compare with [44] since this method utilizes the text extractor to extract the textual description from real images, and then input such text to a diffusion model-based generator to synthesize fake images. Then SVM classifier is trained based on real and such synthesized images. However, we cannot use the same approach to synthesize training samples

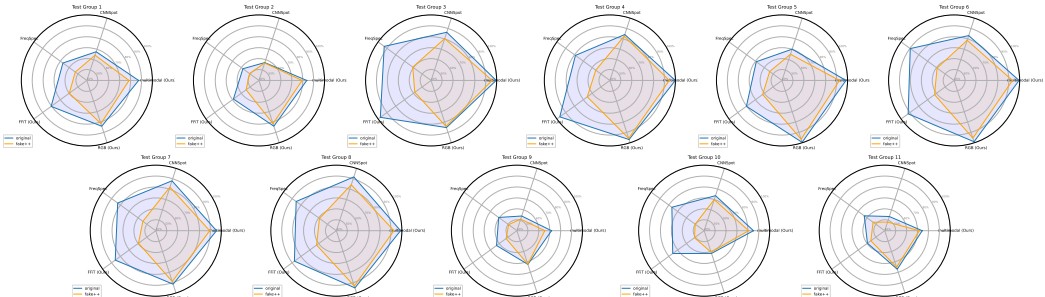

Figure K: Performance evaluation for compromised fake images (evaluated by AUROC).

by Nerf/3DGS methods for group $\mathcal{C}$ and $\mathcal{D}$. We don't compare with [49, 50, 47] since they utilize the inherent features of DM and they are DM-only methods.

## E.1 More Data Samples of NeuroRenderedFake

In Fig. L, we visualize fake image samples generated by exclusive NeRF/3DGS from NeuroRendered-Fake, along with their corresponding camera poses, all belonging to the same 3D scene. Similarly, images samples in Figs. M to Q present fake image samples from the database for the performance evaluation and are generated by a variety of 3D-realistic image synthesis methods, including editable NeRF, editable 3DGS, and combinations of NeRF/3DGS with traditional generative models, extending beyond simple NeRF-based and 3DGS-based approaches. We also give the samples of fake images generated by Sora [22] (test group 12), liveportrait [77] (test group 13) and Runway [21] (test group 14) in Fig. R, Fig. S and Fig. T.

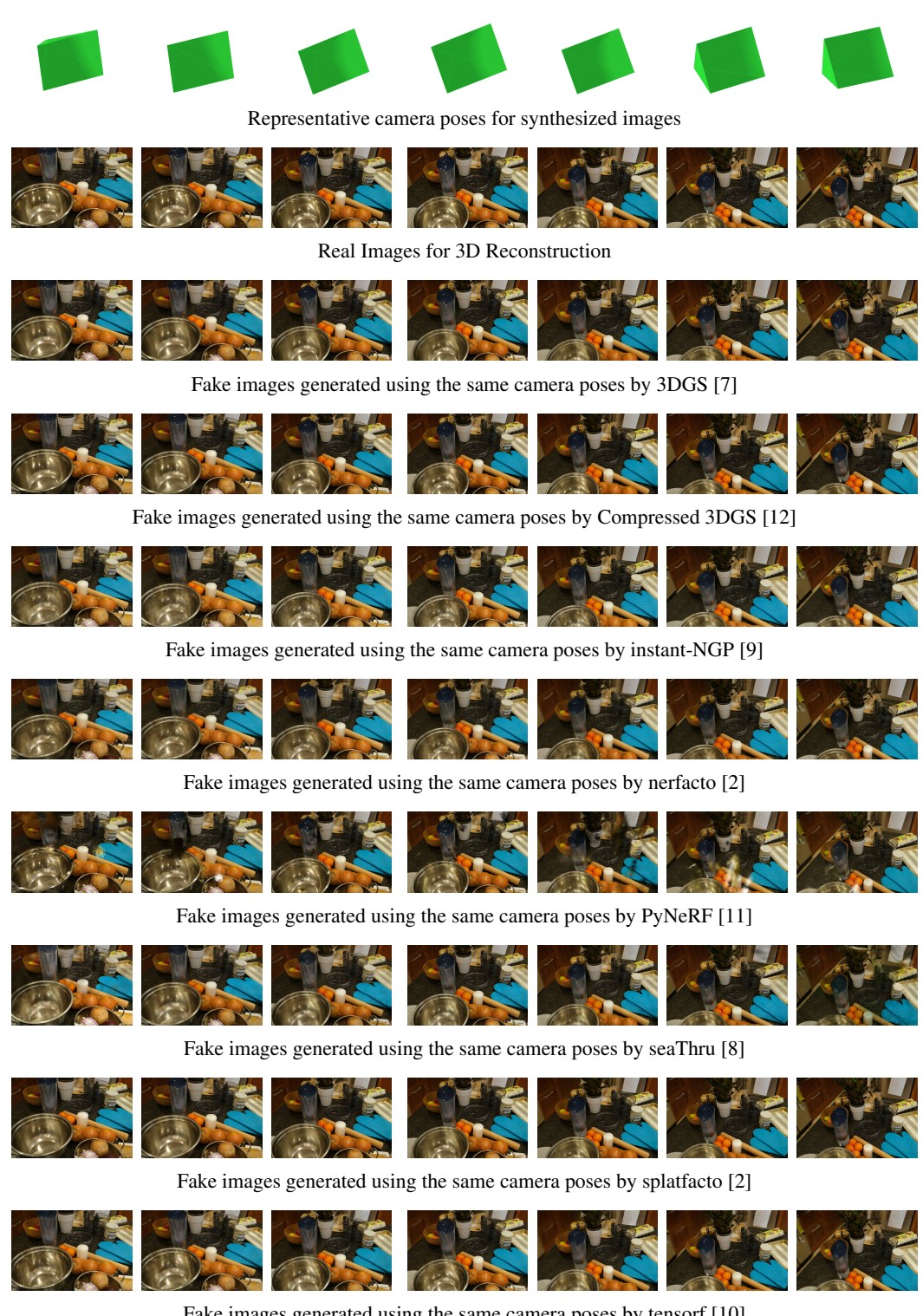

Representative camera poses for synthesized images

Real Images for 3D Reconstruction

Fake images generated using the same camera poses by 3DGS [7]

Fake images generated using the same camera poses by Compressed 3DGS [12]

Fake images generated using the same camera poses by instant-NGP [9]

Fake images generated using the same camera poses by nerfacto [2]

Fake images generated using the same camera poses by PyNeRF [11]

Fake images generated using the same camera poses by seaThru [8]

Fake images generated using the same camera poses by splatfacto [2]

Fake images generated using the same camera poses by tensorf [10]

Figure L: Samples of our acquired dataset for training. The 3D scenes are reconstructed by the different NeRF-based or 3DGS-based methods, from real images with the camera poses.

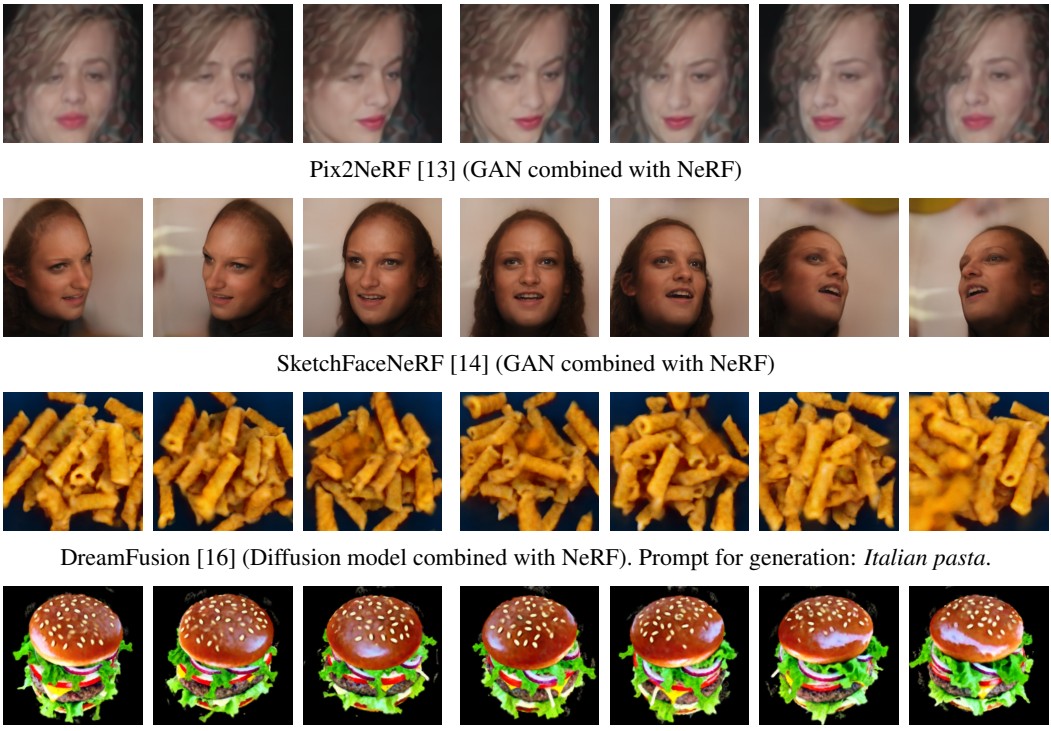

Pix2NeRF [13] (GAN combined with NeRF)

SketchFaceNeRF [14] (GAN combined with NeRF)

DreamFusion [16] (Diffusion model combined with NeRF). Prompt for generation: *Italian pasta*.

GSGEN [17] (Diffusion model combined with 3DGS). Prompt for generation: *American burger*.

Figure M: Samples of fake images synthesized by the methods that combine the generative models with neural rendering technologies.

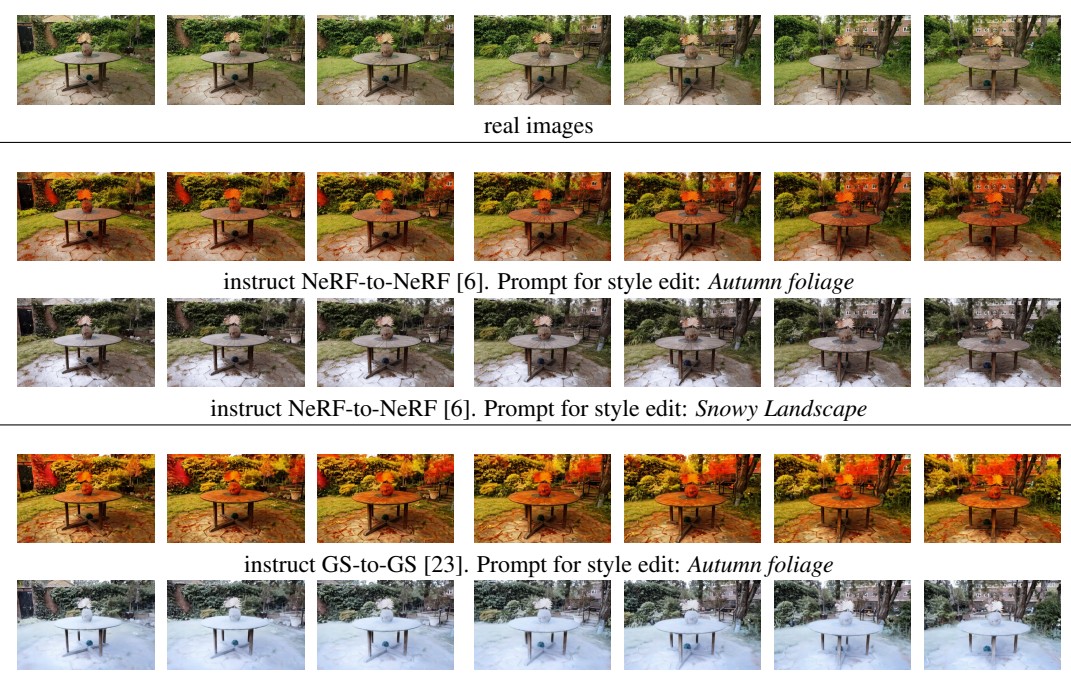

real images

instruct NeRF-to-NeRF [6]. Prompt for style edit: *Autumn foliage*

instruct NeRF-to-NeRF [6]. Prompt for style edit: *Snowy Landscape*

instruct GS-to-GS [23]. Prompt for style edit: *Autumn foliage*

instruct GS-to-GS [23]. Prompt for style edit: *Snowy Landscape*

Figure N: Samples of fake images synthesized by the editable neural rendering technologies.

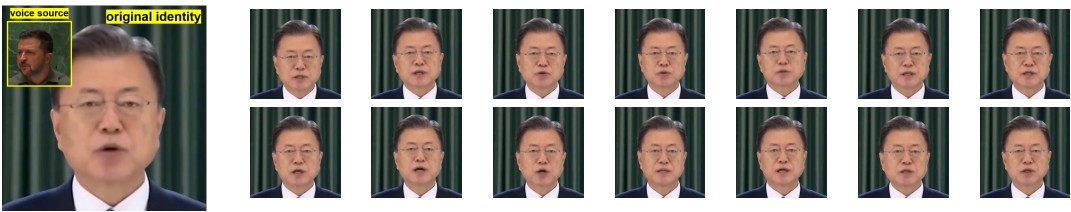

Fake generation process.

Samples of frames extracted from a synthesized fake video, in which the original person's face is manipulated to speak using another person's voice.

Figure O: Generation of the Geneface++ [33] frames for digital avatar synthesis based on NeRF.

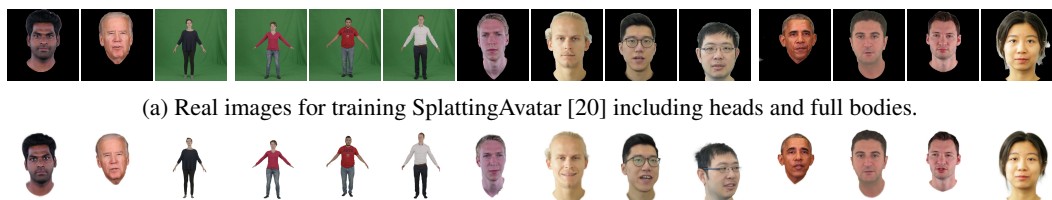

(a) Real images for training SplattingAvatar [20] including heads and full bodies.

(b) Samples of the projected avatars we acquired in the database for each identity.

Figure P: Samples generated by SplattingAvatar [20] which represents neural rendering-based method for digital avatar synthesis.

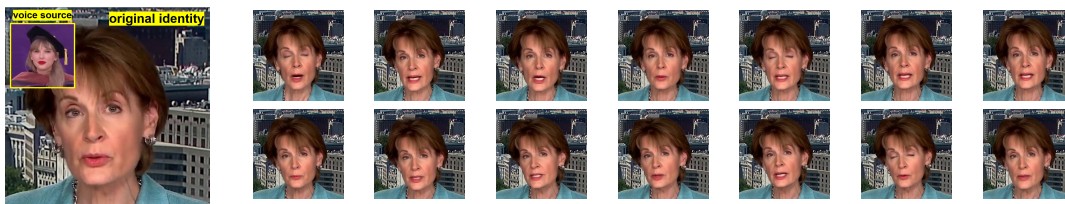

Fake generation process.

Samples of frames extracted from a synthesized fake video, in which the original person's face is manipulated to speak using another person's voice.

Figure Q: Generation of the GaussianTalker [79] frames for digital avatar synthesis based on 3DGS.

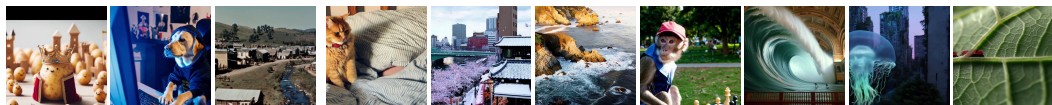

Figure R: Samples of fake images cropped from Sora-generated videos [22].

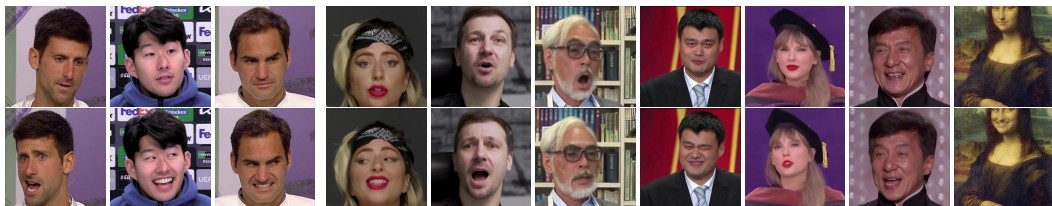

Figure S: Samples of fake images cropped from liveportrait-generated videos [77].

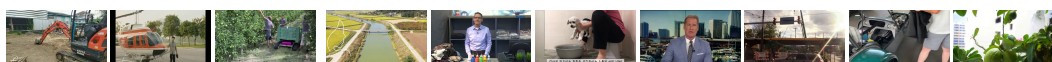

Figure T: Samples of fake images cropped from Runway-generated videos [21].

