# OpenReview forum: "NeuroRenderedFake: A Challenging Benchmark to Detect Fake Images Generated by Advanced Neural Rendering Methods"
_NeurIPS.cc/2025/Datasets_and_Benchmarks_Track — NeurIPS 2025 Datasets and Benchmarks Track poster_

### Official Review · Reviewer_LED5 · 2025-06-29

**Rating:** 4
**Confidence:** 4

**Summary:**

This paper introduces NeuroRenderedFake, a comprehensive benchmark designed to advance fake image detection for images generated by neural rendering. While prior datasets have focused largely on images generated by GANs and diffusion models, this work resolve the growing demand to detect synthetic images created by neural rendering techniques like NeRF and 3DGS. The authors construct a large dataset comprising over 1.6 million neural-rendered images. The paper demonstrates the limitations of existing detectors in recognizing neural-rendered fakes and highlights the value of spectral analysis.

**Dataset Code Accessibility:**

Yes

**Dataset Code Comments:**

The dataset is made available in a well-organized format. The code repository includes implementations of baseline detectors. Expanded tutorials or examples for extending the framework would enhance the accessibility.

**Ethical Considerations:**

No, there are no or only very minor ethics concerns

**Final Justification:**

I have read the rebuttal from the author, which roughly solve my concerns. Although I still worried about the practical usage of the paper, the technical and dataset contribution are valid for me. Therefore, I retain my previous positive rating. My final rating is borderline accept.

**Limitations Weaknesses:**

While the evaluations are thorough within the scope of the benchmark, the paper does not deeply explore adversarial robustness. It remains unclear how well the detectors would handle common adversarial attacks designed to fool detection systems, such as image perturbations, style transfers.

More importantly, unlike compositional fake images generated by GAN like methods, NeRF and 3DGS are mostly used for generating photorealistic novel views of an existing real-world scene. This inherent feature of generation method weakens the practical usage of the dataset: since the generated images are merely reconstruction of the real-world without editing or mis-information, the necessity of detecting them is under question.

**Strengths Contributions:**

A major strength of the paper is that the dataset fills a gap in fake image detection research, which is the detection of neural rendering generated images. Unlike earlier benchmarks that focused on generative model outputs, NeuroRenderedFake addresses the more subtle and realistic outputs of neural rendering pipelines, offering a new resource for the community.

Another strength lies in the spectral energy analysis, where the paper provides a deeper understanding of how frequency domain discrepancies contribute to the results of detection methods. This analytical approach reveals features in frequency domain and support advanced detection model design.

---

> ### Author Rebuttal · Authors · 2025-07-30
>
> ## Response LED5 W1:
> Many thanks for pointing this out. We acknowledge that the current version of the benchmark does not include dedicated adversarial-attack evaluations. The reasons are as follows.
>
> ***Scope of our database-track benchmark submission:*** Previous benchmark papers in this area (e.g., CNNSpot [1], CiFake [67], UniFD [26], GenImage [68], WildFake [78]) likewise concentrate on data collection, protocol design, and baseline corruption robustness (JPEG, Gaussian blur, etc.), and studies developing fake detectors (e.g., spatial-based detectors [1, 33, 34, 70, 71, 72, 80, 82, 83] spectral-based detectors [25, 40, 41, 74, 75]) also primarily evaluate robustness against such common corruptions. We adopted the similar convention and stayed focused so that the evaluation protocol would remains manageable.
>
> ***Future extension:*** We fully agree that detection of adversarial attack is an important and open problem. In the further extension of this work, we intend to evaluate advanced perturbations on benchmark datasets and present comparative results with SOTA methods. We will cite on such extensions and references to adversarial attacks in the final version of this paper.
>
> ## Response LED5 W2:
> As explained in our Related Work section, neural-rendering technologies can be used to generate non-existant scenes, misleading news and social media visuals that go far beyond "plain" view synthesis:
>
> ***Scene editing by prompts (text or image):*** NeRF-based/3DGS-based editors (e.g., Instruct-N2N, instruct-GS2GS) allow arbitrary geometric and photometric alterations of a reconstructed scene from natural-language commands.
>
> ***Unrealistic scene generation:*** Hybrid methods such as GSGEN or DreamGaussian couple 3DGS with diffusion models as backbone and can hallucinate complete scenes that never existed.
> ***Digital-human synthesis:*** Speech-to-video heads (GeneFace++, GaussianTalker) and full-body avatars (SplattingAvatar) produce photorealistic humans that can be driven by arbitrary audio or motion prompts.
> Therefore, these aforementioned cases raise several forensic concerns in uncovering counterfeit visual contents, which the NeuroRenderedFake benchmark is designed to address:
> * Does purely neural-rendered images (NeRF or 3DGS) exhibit latent artifacts that can be detected?
> * When neural rendering is combined with traditional generative models (GAN/diffusion), are those composites still detectable?
> * Can detectors reliably flag popular but high-risk applications such as digital avatars?
>
> By including plain NeRF/3DGS renderings, hybrid NeRF/3DGS + GAN/DM workflows, and avatar-generation tracks in our evaluation groups (please refer to Table 3 and Supp. C.2), the dataset enables a systematic investigation of all three questions.

---

> ### Comment · Area_Chair_4Cym · 2025-08-05
>
> Gentle reminder. Please read through the authors' rebuttal and share any further comments. Thanks!

---

> ### Comment · Reviewer_LED5 · 2025-08-05
>
> I have read the author's response, which generally solves my concerns. Therefore, I would like to keep my original rating.

---

> > ### Author Response · Authors · 2025-08-09
> >
> > We thank the reviewer for reading our rebuttal and for their thoughtful follow-up, which confirms that the comments have been well addressed.

---

### Official Review · Reviewer_JPSi · 2025-07-01

**Rating:** 5
**Confidence:** 5

**Summary:**

The paper releases NeuroRenderedFake, a large-scale image benchmark containing about 0.51M real photos and 1.65M fake images generated by 15 state-of-the-art methods: pure NeRF/3DGS renderers, GAN/DM-to-NeRF hybrids, and neural-rendered digital avatars.

In this benchmark, three evaluation tasks are proposed:
1) Cross-domain: train on one synthesis family (GAN, DM, NeRF, 3DGS) and test on another;

2) Cross-time: examine whether detectors trained on older renderers generalise to newer ones; and

3) Degradation robustness: measure the impact of JPEG and blur corruptions.

Besides, a two-branch multimodal detector , which contains ViT-L RGB and ViT-B FFT fused via a gated unit, is introduced, outperforming re-implemented spatial, spectral, and FatFormer baselines on most cross-domain splits.

**Dataset Code Accessibility:**

Yes

**Ethical Considerations:**

No, there are no or only very minor ethics concerns

**Final Justification:**

Thanks for the authors' response. The dataset itself is strong enough as a valid contribution to this field, and thus, I raise my rating to 5.

**Limitations Weaknesses:**

1. Methodological novelty is incremental. The detector is a straightforward two-branch ViT with late gating, while similar RGB + FFT designs already exist.

2. FatFormer’s language branch is removed for the claimed fairness, yet language could aid cross-domain generalisation. Therefore, including the original FatFormer or another LMM-based detector as an upper bound might be more fair to the method itself, or justify removal more rigorously.

3. Minor issue: pick one spelling throughout--``NeuroRenderedFake'' vs ``NeuroRenderedFakes''

**Strengths Contributions:**

1. The coverage of neural rendering. Prior datasets focus on GAN/DM imagery, while this work adds NeRF/3DGS and hybrid artefacts at scale.

2. Insightful spectral-energy study. Weight statistics quantify when FFT features help or hurt, which is more helpful than treating multimodal as a black box。

3. Well-designed evaluation protocols (cross-domain, cross-time, degradation) reveal weaknesses that IID testing hides.

---

> ### Author Rebuttal · Authors · 2025-07-30
>
> ## Response JPSi W1:
> We thank reviewers for insightful comments. As also addressed in our response to **Reviewer yMwc W2**, existing SOTA multimodal detectors (e.g., FatFormer) typically employ early fusion, where spatial and frequency features are merged at an early stage, thus preventing any clear quantification of each domain’s contribution to the final decision.
>
> To address this, we propose a simple architecture in which the two modalities are kept strictly separate throughout the network and are fused only at the final layer via a single gated-multimodal unit. The learnable weight $w_b$ in this unit explicitly quantifies the relative influence of the spectral domain. This design is not aimed at achieving superior detection performance, but rather at enabling a controlled, interpretable analysis of:
> * How different training protocols affect the model’s reliance on spatial vs spectral artifacts;
> * How the two domains behave across the diverse synthetic image types in NeuroRenderedFake dataset.
>
> In this context, the detector serves primarily as an analytical tool for our benchmark dataset rather than as an independent contribution, which is why it is not emphasized among the paper’s key contributions.
>
> ## Response JPSi W2:
> Why we retain only the vision branches and what the proposed multimodal backbone is meant to show:
>
> ***Purpose of the backbone:*** Improving raw detection accuracy is not the main goal of this benchmark paper. A multimodal architecture is included solely to:
> * Quantify, via the weight $w_b$, that how much spatial versus spectral evidence contributes to the final decision respectively
> * Uncover which of the two domains is more robust under the diverse training/testing configurations provided by NeuroRenderedFake.
>
> ***Excluding language-model branches:*** • GAN/DM fakes are not rendered from a real scene, therefore, they may contain hallucinations [R1] that do not match the reality, making them detectable by an LLM from spatial-logic inconsistencies. However, NeRF/3DGS renderings produce more realistic synthetic images by reconstructing scenes from actual images, thereby avoiding logical and semantic inconsistencies. • Including text features can therefore give an uneven advantage to the GAN/DM part of the benchmark and blur the comparison we wish to make between classic generative fakes and neural-rendered fakes. • To keep the evaluation fair and focused on low-level artefacts, we disable all language branches, treating every baseline as vision-only.
>
> ***Potential for Future Development of Detectors:*** As shown from the FatFormer ablation study, the quality of the vision backbone remains a significant factor, even with the assistance of language-guided information. We believe that the developed multimodal architecture, with separate branches representing two distinct domains, can serve as a vision backbone for designing more effective fake image detectors in the future.
>
> In short, the backbone is primarily an analytic tool, and omitting language features ensures a balanced, artifact-specific comparison across the very different kinds of synthetic images in our benchmark dataset.
>
> ## Response JPSi W3:
> Thanks for pointing out this spelling error, we will surely fix this error in the final version of our paper.
>
> ### References
>
> [R1] Aithal, Sumukh K., et al. "Understanding hallucinations in diffusion models through mode interpolation." Advances in Neural Information Processing Systems 37 (2024): 134614-134644.

---

> ### Comment · Area_Chair_4Cym · 2025-08-05
>
> Gentle reminder. Please read through the authors' rebuttal and share any further comments. Thanks!

---

### Official Review · Reviewer_yMwc · 2025-07-02

**Rating:** 4
**Confidence:** 3

**Summary:**

The paper introduces a dataset consisting of a curation of existing datasets + a large addition of fake images from neural rendering approaches and GS, to be used for fake image detection and benchmarking. A protocol to benchmark detectors is provided, which also evaluates the impact of training on specific image generation pipelines vs testing on others, and multiple evaluations of existing methods are provided. A new method for fake image detection is introduced, with minor improvements over existing methods.

**Dataset Code Accessibility:**

Yes

**Dataset Code Comments:**

Code and data are accessible, but seem not very well organized (see above). I would advise the authors to improve the repositories.

**Ethical Considerations:**

No, there are no or only very minor ethics concerns

**Final Justification:**

The authors addressed my concerns. I believe this paper provides a good contribution, but I am not satisfied with its presentation/clarity. I think the paper should be accepted, conditionally to improving such aspects for the final version, but I would not oppose its rejection.

**Limitations Weaknesses:**

W1) I had difficulty in understanding the structure of the dataset. The tables in the paper help a bit, but it is not immediately clear what the dataset contains, how it is divided and why. Moreover, the repository seems to be fairly unstructured, making it very unclear which parts of the dataset have to be used in their respective cases. Relating to unclarity, could the authors clarify the meaning of "FFiT", which is used to refer to the frequency branch of their architecture? It is mentioned but never explained.
W2) Lines 66-71 include a claim about a new fake detection architecture, however this claims is not listed in the contributions of the paper right below it. It is unclear whether this is a new architecture and whether it is in the main purposes of the paper to describe and evaluate it.
W3) The paper considers neurally (or GS) rendered images from a pose existing in the training set to be fake (figure 1). In my opinion, these images are not fake, because the models have been trained to reproduce them correctly. INR-based image-compression models work in a similar way, so I view these images as simply "compressed" through a neural network or a set of Gaussians. Instead, an image generated from a slightly different camera pose should be considered fake, because that camera pose did not exist in reality. Could the authors elaborate on this?
W4) Lines 260-262 analize how image quality affects fake detection, and conclude that JPEG is a greater challenge than blurring. This doesn't seem to backed by the results (figure 5), which seem to present a mixed situation.

**Strengths Contributions:**

S1) I believe in the importance of such dataset, allowing the training and evaluation of fake image detectors not only for GAN and diffusion models but also for NVS models.
S2) The dataset is large and contains images generated with multiple methods
S3) The frequency analysis is interesting: it shows that some image generation methods exhibit a bias in the frequency domain, which can be exploited by fake detectors, but not all methods exhibit the same bias. Future work can target this direction.

---

> ### Author Rebuttal · Authors · 2025-07-30
>
> ## Response yMwc W1:
> We thankfully appreciate reviewer’s comments. Below (1) we provide a concise description of what the NeuroRenderedFake dataset contains, how it is organized, and why, (2) explain the directory layout that appears in the public release, and (3) explain the term "FFiT".
>
> **(1) Composition, organization, and rationale of the NeuroRenderedFake dataset**
>
> **Training protocol:** In Table 2 of the main text, the rows labeled **A**-**O** represent the image sources used to construct the 3D scenes. We list these sources to ensure that the construction of the database can be fully replicated by other researchers and we confirm that all image sources are publicly accessible. In the Supp. C.1, we describe the training groups. Groups $\mathcal{A}$ and $\mathcal{B}$ focus on GAN- and diffusion-based fake images and therefore are not directly related to the neural-rendered scenes listed in Table 2. Group $\mathcal{C}$ uses all real images from sources **A**-**J** and their rendered class produced by methods I-V, whereas Group $\mathcal{D}$ pairs the same real images with every 3DGS render from sources **A**-**J**. By specifying the source scenes and rendering methods, we enable full reimplementation of the benchmark (e.g., **D**-Ⅱ denotes Tensorf renders of the mill19 dataset). We train only on scenes **A**-**J** and reserve **K**-**O** to evaluate how well NeRF/3DGS detectors generalize to unseen neural-rendered fakes.
>
> **Evaluation protocol:** For Table 3 in the main text, in order to evaluate detectors trained on GAN/DM/NeRF/3DGS and tested on unseen NeRF/3DGS, we split the scenes into two parts: **A**–**J** and **K**–**O**. The images from **K**–**O** are never exposed to the detectors during training, which uses data from **A**–**J** as training set. As a result, the unseen NeRF-rendered images form testing group 1, and the unseen 3DGS-rendered images form testing group 2. Additionally, the synthetic images from **K**–**O** are basic fake images generated by neural rendering methods. We further extend the fake detection task to more complex cases beyond simple neural rendering, including: editable NeRF (e.g., Instruct-N2N), editable 3DGS (e.g., Instruct-GS2GS), hybrid methods with diffusion or GAN methods (e.g., GSGEN, DreamGaussian), 3DGS-based avatars (e.g., GeneFace++, GaussianTalker, SplattingAvatar), and realistic video frames. These more advanced cases constitute testing groups 3 to 14. In Supp. C.2, we also provide detailed statistics of our database, including the number of real and fake images in each directory of the released dataset.
>
> **(2) Explanation about the released repository for the database**
>
> Due to Harvard Dataverse's single file size limits, it is not possible to upload our material using the nested directory hierarchy originally designed. Furthermore, since our benchmark database exceeds 300 GB, split compression is necessary to comply with the platform's storage restrictions. As a result, the current public release consists of multiple compressed archives, and their placement within the Dataverse directory tree does not directly reflect the intended train/test protocol splits. Consequently, the full structure of the database can be properly visualized when reviewers download and extract all the files.
>
> To avoid ambiguity, our Dataverse record also provides a comprehensive README.md that enumerates every archive and explains which archives correspond to the four training pools ($\mathcal{A}$–$\mathcal{D}$) and to each evaluation group (1–14), as well as the details of the numbers of real/fake images in each folder. More details for each training group appear in Supp. C.1, and those for evaluation groups in Supp. C.2.
>
> Furthermore, in the code repository URL we have provided a Python file named my_dataset.py, which serves as the dataloader for both training and testing. During training, any user can specify the desired training group by setting the argument --training_group='A' (or 'B', 'C', 'D'), and the data loader will automatically search for and load all training images from the corresponding group in the provided database. Similarly, for performance evaluation on the test set, user can set --testing_group='1' (or '2', ..., '14') to load all images from the specified testing group. If the paper is accepted, we will also mirror the dataset on our lab server (HTTP+rsync) and on Google Drive, preserving the original directory structure for a more readable layout.
>
> **(3) The term FFiT**
>
> We denote the frequency branch of our architecture as the **F**ourier-**F**requency-**i**nformed **T**ransformer (FFiT). This module is based on a ViT-B/16 backbone that is pre-trained exclusively on the 2D Fourier spectrum of natural images (see Supp. A.2 for details). The pre-training is performed using masked auto-encoding with a focal–centrosymmetric loss. The purpose of FFiT is to model frequency-domain characteristics and serve as the spectral branch in our multimodal detection framework. We will surely clarify this abbreviation "**F**ourier-**F**requency-**i**nformed **T**ransformer (FFiT)" for the first time (at L166-168) the term appears in Sec. 4.1 and reference to Supp. A for training details.
>
> ## Response yMwc W2:
> Our primary contribution is the NeuroRenderedFake benchmark, which comprises (1) the released database and (2) evaluation protocol; Please note that the third contribution (3) mentioned in this paper is ***'Analysis of Spectral Energy Distribution'***, and the proposed multimodal detector is an auxiliary tool that supports this last contribution.
>
> SOTA detectors like FatFormer fuse spatial and frequency features early, obscuring each domain's contribution to the final decision. Yet our cross-domain analysis requires precisely that diagnosis to enhance understanding of such contributions, as it is vital to design the next generation of detectors. We therefore introduce a simple architecture that keeps the two modalities strictly separated and only fuse their features in the last stage: the branches are combined only through a single gated-multimodal unit at the last layer, whose learnable weight $w_b$ explicitly indicates the relative contribution of the spectral domain.
> The goal of this design is clearly *not to* achieve a new SOTA detection accuracy, but to enable a controlled, quantitative assessment of:
> * How the reliance (inherent bias for the multimodal architecture) on spatial vs. spectral artifacts changes under different training protocols: Please refer to L271–287 for detailed descriptions of the statistical trends in the distribution of the weight $w_b$, which represents the bias learned by the multimodal model under different training conditions.
> * How the two domains behave for the diverse kinds of synthetic images contained in NeuroRenderedFake (plain NeRF, 3DGS, NeRF + DM, editable 3D avatars, etc.): Please refer to Supp. B.2 and Figure J of the violin plots for the distributions of $w_b$ across different testing groups.
>
> The detector thus serves as an analysis tool for the benchmark, and that is why we did not list it among the headline or the key contributions of the paper.
>
> ## Response yMwc W3:
> We understand the reviewer’s concern and agree that the terminology can be confusing. We clarify this aspect in the following.
>
> ***Definition used in the paper:*** We label as fake any image that is numerically generated by a NeRF or 3DGS renderer and therefore bears renderer-specific artifacts, regardless of whether the viewpoint existed in the original capture set. We will ensure this is explicitly stated in the final version *(fake = generated image that contains specific artifacts introduced by a neural renderer)*.
>
> ***Why “plain” neural rendered fakes are required:*** The main objective of the benchmark is to study whether the artifacts found in simple, un-edited renderings also appear in more complex outputs:
> * editable NeRF (e.g., Instruct-N2N)
> * editable 3-DGS (e.g., Instruct-GS2GS)
> * hybrids with DM/GAN backbones (e.g., GSGEN, DreamGaussian)
> * 3DGS avatars (e.g., GeneFace++, GaussianTalker, SplattingAvatar).
>
> To answer this question, detectors must first learn the artifact signature in plain NeRF and 3DGS images. Our experiments show that models trained on groups $\mathcal{C}$ and $\mathcal{D}$ (plain NeRF/3DGS) generalize well to the more complicated groups 3–14, confirming that those specific artifacts persist. As highlighted in our response to **Reviewer LED5 W2**, it is important to emphasize that the use of plain, neural-rendered fake images plays a critical role in our analysis. These seemingly simple synthetic images provide a clean and controlled environment for isolating and studying the inherent artifacts that arise in neural rendering. Such artifacts are not only characteristic of basic neural rendering techniques but are also consistently present across more complex and advanced synthetic image generation methods. By analyzing these fundamental cases, we gain valuable insights that generalize to a broader range of synthetic content.
>
> ***Future plan:*** We plan to enlarge the dataset with additional unseen-view images to enhance the diversity of our database.
>
> ## Response yMwc W4:
> Thank you for this important observation. We agree that our original phrasing was not precise as it missed citing "spectral branch". Our intended point was to highlight that JPEG compression imposes a particularly strong challenge to the *spectral branch* of our detector. This is intuitive, as JPEG compression introduces characteristic artifacts in the frequency domain, which interfere with the spectral features the model relies on (i.e., the magnitude spectrum) for fake detection. As a result, distinguishing real from fake images becomes notably harder under JPEG compression, particularly for the frequency-based branch. *We will clarify in L260–262 of the final manuscript that it is the "spectral branch" that is more vulnerable to noise from JPEG compression.*

---

> > ### Comment · Reviewer_yMwc · 2025-08-05
> >
> > I thank the authors for the detailed response. I understand their point of view, my criticisms have been addressed. I kindly ask them, in case of acceptance, to refine a bit the description of the dataset structure. It has been clarified in the rebuttal and I believe the main paper can benefit on additional clarity in this regard.

---

> > > ### Author Response · Authors · 2025-08-09
> > >
> > > We thankfully appreciate reviewer’s comments and time in reading our clarifications. We will definitely make minor edits to incorporate reviewers’ suggestions in the camera-ready version of this paper.

---

> ### Comment · Area_Chair_4Cym · 2025-08-05
>
> Gentle reminder. Please read through the authors' rebuttal and share any further comments. Thanks!

---

### Decision · Program_Chairs · 2025-09-18

**Decision:**

Accept (poster)

**Comment:**

(a) Summary of Scientific Claims and Findings

This paper presents NeuroRenderedFake, a benchmark and dataset for detecting fake images generated by neural rendering techniques such as NeRF and 3D Gaussian Splatting. It includes: (1) a large-scale dataset spanning plain to hybrid neural-rendered fakes, (2) a cross-domain evaluation protocol, and (3) spectral analysis using a diagnostic multimodal detector to study spatial vs. frequency artifact reliance.

The paper makes three primary contributions:
* A large-scale dataset of synthetic images generated using state-of-the-art neural rendering techniques, covering both "plain" and hybrid/edited outputs.
* A cross-domain evaluation protocol that includes 15 groups of testing images to assess generalization from neural-rendered fakes to more complex compositions (e.g., avatars, hybrids with GANs/DMs).
* A spectral analysis framework, supported by a lightweight multimodal detector that separately models spatial and frequency information, enabling interpretation of detector biases across training/testing domains.

(b) Strengths
* Tackles an underexplored yet growing threat: neural-rendered fakes.
* Well-structured evaluation protocol: The training/testing splits are carefully designed to test generalization across synthetic modalities, with real use cases such as text-driven editing and avatar generation well-represented.

(c) Weaknesses and Limitations
* Lacks adversarial robustness evaluation, but this is beyond the expected scope for DB Track and consistent with prior benchmarks.
* Some terminology ambiguity was initially noted (e.g., the definition of "fake" images), and while the rebuttal clarified this as renderer-generated content with observable artifacts, this should be made explicit in the final version.
* The multimodal detector caused mild confusion among reviewers about its role. The authors clarified it is not a main contribution but rather a tool for protocol analysis.

(d) Final Recommendation and Justification

I recommend acceptance of this paper in the DB Track. It delivers a substantial and well-justified benchmark,, filling a crucial gap by targeting neural-rendered synthetic media. The work stands out not because of flashy results, but because it provides a high-utility resource, structured in a reproducible way and supported by thoughtful protocol design. The inclusion of complex generation methods (e.g., NeRF+DM hybrids, avatars) and focus on artifact-level generalization address real-world forensic concerns.

(e) Rebuttal Summary and Discussion

Several key points were raised during the review and rebuttal process:
* Terminology around "fake" and neural rendering:
Concerns: Lack of clarity about whether images synthesized from real captures should be labeled as "fake".
Response: The authors clarified that any image synthesized via a renderer—regardless of view novelty—is labeled "ㄹake" due to inherent artifacts. This is now slated for explicit mention in the revised paper.
Assessment: Accepted clarification, in line with DB Track's emphasis on clear, operational definitions.
* Role of the detector and contribution framing:
Concerns: Whether the inclusion of the multimodal detector constitutes a method contribution or is outside scope.
Response: Authors explained that the detector serves solely to analyze spatial vs. spectral artifact reliance under different training conditions. This is not emphasized as a standalone method contribution.
Assessment: Clear alignment with DB Track's emphasis on analysis and evaluation protocols.
* Missing adversarial attack evaluation:
Concerns: One reviewer noted that the benchmark omits adversarial robustness, which may limit practical impact.
Response: Authors acknowledged this as important future work and noted that adversarial attacks are not commonly included in similar prior benchmark papers.
Assessment: Reasonable trade-off; not essential for DB Track acceptance given focus on data and evaluation.
* Suggestions to improve clarity in final version:
Requests: Minor edits to improve dataset structure explanation and clearer introduction of terms like FFiT.
Response: Authors agreed and will update the final manuscript accordingly.
Assessment: Positive attitude and adequate response.

Overall, the rebuttal period improved the clarity and completeness of the submission. The authors were constructive and thorough in addressing all points raised.

Final Verdict: Accept (DB Track)
The paper is well-scoped, clearly within the DB Track CFP, and offers a meaningful benchmark contribution with both practical utility and diagnostic depth.